



# Canopy Area of Large Trees Explains Aboveground Biomass Variations across Nine Neotropical Forest Landscapes

Victoria Meyer[1,2], Sassan Saatchi[1], David B. Clark[3], Michael Keller[4,5], Grégoire Vincent[6], António Ferraz[1], Fernando Espírito-Santo[1,7], Marcus V.N. d'Oliveira[5], Dahlia Kaki[1] and Jérôme Chave[2]

[1] *Jet Propulsion Laboratory, California Institute of Technology, Pasadena, CA. USA*
[2] *Laboratoire Evolution et Diversité Biologique UMR 5174, CNRS Université Paul Sabatier, Toulouse, France*
[3] *Department of Biology, University of Missouri, St. Louis, Missouri, U.S.A.*
[4] *USDA Forest Service, International Institute of Tropical Forestry, San Juan, Puerto Rico*
[5] *EMBRAPA Acre, Rio Branco, Brazil*
[6] *IRD, UMR AMAP, Montpellier, 34000 France*
[7] *Lancaster Environmental Centre, Lancaster University, Lancaster, United Kingdom, LA1 4YQ*

*Correspondence to:*

Victoria Meyer
*Jet Propulsion Laboratory*
*California Institute of Technology*
*4800 Oak Grove Drive*
*Pasadena, CA. 91109 USA*
*Email: victoria.meyer@jpl.nasa.com*



## Abstract

Large tropical trees store significant amounts of carbon in woody components and their
distribution plays an important role in forest carbon stocks and dynamics. Here, we explore the
properties of a new Lidar derived index, large tree canopy area (LCA) defined as the area
occupied by canopy above a reference height.  We hypothesize that this simple measure of forest
structure representing the crown area of large canopy trees could consistently explain the
landscape variations of forest volume and aboveground biomass (AGB) across a range of climate
and edaphic conditions. To test this hypothesis, we assembled a unique dataset of high-resolution
airborne Light Detection and Ranging (Lidar) and ground inventory data in nine undisturbed old
growth Neotropical forests.  We found that the LCA for trees greater than 27 m (~25–30 m) in
height and at least 100 m$^2$ crown size in a unit area (1 ha), explains more than 75 % of total
forest volume variations, irrespective of the forest biogeographic conditions. When weighted by
average wood density of the stand, LCA can be used as an unbiased estimator of AGB across all
sites ($R^2$ = 0.78, RMSE = 46.02 Mg ha$^{-1}$, bias = 0.76  Mg ha$^{-1}$). Unlike other Lidar derived
metrics with complex nonlinear relations to biomass, the relationship between LCA and AGB is
linear.  A comparison with tree inventories across the study sites indicates that LCA correlates
best with the crown area (or basal area) of trees with diameter >50 cm.  The spatial invariance of
the LCA–AGB relationship across the Neotropics suggests a remarkable regularity of forest
structure across the landscape and a new technique for systematic monitoring of large trees for
their contribution to AGB and changes associated with selective logging, tree mortality, and
other types of forest disturbance and dynamics.

## Keywords


Lidar, biomass, tropical forest, large trees, crown area, wood density



## 1    Introduction

In humid tropical forests, tree canopies contribute disproportionately to the exchange of water and carbon with the atmosphere through photosynthesis (Goldstein et al., 1998; Santiago et al., 2004). From a physical standpoint, canopies are rough interfaces formed by crowns of emergent and large trees, regularly disturbed by wind thrusts and gap dynamics. This structurally complex boundary layer is challenging for scaling of biogeochemical fluxes and modeling of vegetation dynamics (Baldocchi et al., 2003). Large canopy trees are among the first to be impacted by storms or heavy precipitation (Espírito-Santo et al., 2010), drought stress (Nepstad et al., 2007; Saatchi et al., 2013; Phillips et al., 2009), and fragmentation (Laurance et al., 2000), potentially leading to tree death and formation of large canopy gaps (Denslow, 1980; Espírito-Santo et al., 2014). Several studies suggest that forest canopies can show fractal properties that tend to evolve from a non-equilibrium state towards a self-organized critical state, involving gap formation and recovery (Pascual and Guichard, 2005; Solé and Manrubia, 1995), with crowns preferentially growing towards more sunlit parts of the canopy (Strigul et al., 2008).

Over the past decade, stand level canopy metrics have been increasingly derived using small footprint airborne Lidar systems (ALS), a widely used remote sensing technique to study the structure of forests (Kellner and Asner, 2009; Lefsky et al., 2002). Lidar derived mean canopy height (MCH) is a good predictor of tropical forest aboveground carbon content and its spatial variability (Jubanski et al., 2013), but it does not provide information on the presence of large trees that are important when monitoring changes of forest biomass from logging and small scale disturbance (Bastin et al., 2015). Moreover, different forests with the same MCH may differ in their stem density, notably of large trees, and in stand mean wood density, two aspects that are important in constructing a robust model to infer AGB from lidar data (Asner et al., 2012;



Mascaro et al., 2011). Ground observations suggest that stem density, basal area, height and
crown size of large tropical trees may all be good indicators of forest AGB (Clark and Clark,
1996; Goodman et al., 2014). This implies that including information on crown area of
individual large trees should improve carbon stock assessments, as confirmed in temperate and
boreal regions (eg. Packalen et al., 2015; Popescu et al., 2003; Vauhkonen et al., 2011, 2014).  In
tropical forests, identifying and delineating crowns of large trees is a difficult and time
consuming process due to the layered structure of the forest canopy and overlapping crowns
(Zhou et al., 2010, but see Ferraz et al., 2016).
Here, we explore how the fractional area occupied by crowns of large trees in a forest stand can
be used as a reliable indicator of forest biomass across a wide range of forest structure, climate
and edaphic geographic variations.  We define large tree canopy area (LCA) as a metric
capturing the cluster of crowns of large trees within a forest patch using height and crown area
measured by high resolution airborne Lidar measurements. Precisely, LCA is the number of
pixels in the canopy height model above a reference height, and excluding the pixel clusters
smaller than a reference area. Since this metric quantifies the proportional presence of large
trees, it can be used to estimate AGB and monitor changes associated with the disturbance of
large trees from mortality events and selective logging.   We first explore the properties of LCA
across a range of landscapes in the Neotropics. Next, we hypothesize that LCA is a good
predictive metric of the spatial variations of AGB over a wide range of old growth forests.
To this end, we assembled a collection of airborne Lidar measurements and ground inventory
data at nine sites in old growth Neotropical forests. The Lidar data provide variations in canopy
height and distribution of large trees that allow us to address the following questions: 1) is there





a unique definition of LCA at the landscape scale across different sites? 2) does LCA metric
capture variations of AGB?

## 2    Materials and Methods

### 2.1    Study sites

We studied the canopy structure at nine old growth lowland Neotropical forest sites that span a
broad range of climatic and edaphic conditions (Fig. S1, Table 1). All sites are located in low
elevation areas (less than 500 m above sea level) but have small scale surface topography that
may influence the distribution of crown formations and gaps. These forests are for the most part
undisturbed *terra firme* forests. Tapajós, Antimary and Cotriguaçu get the least rainfall, with
approximately 2000mm yr$^{-1}$, while La Selva and Chocó both receive more than 4000 mm yr$^{-1}$
(Table 1).
Permanent forest inventory plots were available for all sites except Cotriguaçu (Table 1). Sites
where tree level inventory data were available were used to estimate the stand level aboveground
biomass, thereafter referred to as AGB$_{inv}$: BCI (50 plots of 1 ha each), Chocó (42 plots of 0.25 ha
each), La Selva (11 plots of 1 ha each), Manaus (10 plots of 0.25 ha each), Nouragues (7 plots of
1 ha each) and Tapajós (10 plots of 0.25 ha each). In these plots, all trees with a diameter at
breast height (DBH) ≥10 cm have been mapped, measured and identified to the species. Trees
with irregularities or buttresses were measured higher on the bole. Total tree height
measurements were available for a subset of these trees. The method for calculating AGB$_{inv}$ from
forest inventories at 1 ha scale is reported in S.1 of the supplementary information. Stand
averaged wood density of each site was calculated and is reported in Table 1. Additional plot



level data (AGB$_{inv}$ and mean wood density) were provided for Antimary (50 plots of 0.25 ha
each), Nouragues (27 plots of 1 ha each) and Paracou (85 plots of 1 ha each).
The four sites where 1 ha plots were available were used to compare the LCA metric and AGB,
and are here referred to as "calibration sites" (BCI, La Selva, Nouragues and Paracou). Smaller
plots have a higher probability of having the crown of large trees extend outside the plot
boundary, which can introduce uncertainty in estimates of LCA because of edge effect (Meyer et
al., 2013; Packalen et al., 2015). For this reason, all plots smaller than 1 ha were excluded from
this analysis.

**2.2    Lidar data**
Lidar sensors scan the vegetation vertical structure and return a three dimensional point cloud
derived from the time it took each pulse to return to the instrument. The Lidar datasets acquired
over the study sites come from discrete return Lidar instruments and were gridded horizontally at
a 1m resolution using the echoes classified as either vegetation or ground. They yield three
products: digital surface model (DSM) corresponding to the top canopy elevation, digital terrain
model (DTM) corresponding to the ground elevation, and canopy height model (CHM), which is
the height difference between the DSM and the DTM. DTMs were interpolated from a Delaunay
triangulation or comparable interpolation methods, after outliers have been removed. DSMs were
created using the highest return within a cell. Lidar data over Paracou were acquired in last
return mode, causing a bias of 50 cm on the CHM (Vincent et al., 2012). This bias is not
addressed in this study because our height increment for the determination of optimal height
thresholding is larger (1m) (see Sect. 4.3). Data were acquired between 2009 and 2013, using



relatively similar sensors and acquisition configurations (Table 2). The potential differences
between the Lidar datasets and their impact on the results are addressed in the Discussion.
For each site, we selected a 1x1 km (100 ha) area of old growth forest, oriented north-south,
without any human disturbance to the extent possible. Topography derived from Lidar data
within the selected 1 km$^2$ subset images provides information on landscape variations that may
impact the forest structure. Data visualization was done using ENVI version 4.8 (Exelis).
Mean canopy height (MCH) is a good predictor of AGB provided that the regression model is
calibrated locally. It was calculated by averaging all the canopy height model pixels falling in an
area of interest. Here, we calculated an AGB map of each site from MCH using the following
model form (Eq. (1), Asner and Mascaro, 2014).
$$AGB_{Lidar} = aMCH^b + \epsilon \qquad\qquad (1)$$
where $AGB_{Lidar}$ is the aboveground biomass estimation derived from Lidar data, $a$ is a scaling
constant, which is expected to depend significantly on forest type and stand level wood density,
$b$ is a power law exponent and $\epsilon \sim N(0, \sigma^2)$ represents the uncertainty in measurements. All
coefficients are presented in Table S1. We inferred the model parameters directly for the sites
where $AGB_{inv}$ of 1 ha plots was available  (La Selva, BCI, Paracou and Nouragues).  For Chocó
and Antimary, we developed models based on 0.25 ha plots and 50 m x 50 m pixels of Lidar data
and after estimating $AGB_{Lidar}$, aggregated the image to 1 ha or 100 m pixels.  For the remaining
sites of the Central Amazon (Cotriguaçu, Manaus and Tapajós), we used a model based on
existing data derived from airborne and spaceborne Lidar (Lefsky et al., 2007).  This model may
have larger uncertainty in estimating biomass compared to our site specific model, but we here
assume that all 1 ha scale $AGB_{Lidar}$ estimates have approximately similar uncertainties.


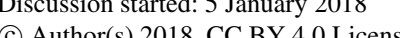


**2.3   Computing Large Canopy Area (LCA)**
At each study site, we extracted the area of canopy that relates to total area of the canopy height
model above a standard height (h) threshold, or LCA(h), and explored how this metric scales
along two axes. First, we varied the threshold height h with increments of 1m, between 5m and
50m, in 100 m by 100 m subareas (100 subareas for each site).  Second, to denoise the data, we
excluded the clusters with less than a set number of $1m^2$ pixels (50, 100, 150 or 200). We then
prioritized the crown area of large trees, and filtered out pixels that could be related to outliers or
to single branches. This method thus quantifies the area of large crowns covering a plot or larger
landscape unit area, as a percentage of covered area.
LCA maps were produced at 1 ha resolution. Pixel clustering was based on the similarity of the
four nearest neighbors (similar results were obtained with an eight neighbor model, results not
shown here). Figure S2 summarizes the steps taken to go from the Lidar canopy height model to
the final LCA map.  Processing was conducted using the IDL software (Interface Description
Language, Exelis).
We determined the optimal minimum crown size and canopy height threshold calculating the
coefficient of correlation between $AGB_{Lidar}$ and LCA. We also performed the same analysis
using $AGB_{inv}$ and LCA at the four calibration sites. This step allowed us to examine if optimal
height thresholds differed from one site to the other. The goal was to find a single optimal height
threshold and crown size that could be applied for LCA retrieval across closed canopy
Neotropical forests.

**2.4   Relating LCA to biomass**



We tested different models to infer $AGB_{inv}$ from LCA, henceforth called $AGB_{LCA}$, at the four
calibration sites, and explored if adding more parameters, such as mean wood density of a site,
mean wood density of large trees (DBH $\geq$50 cm), mean canopy height or top percentiles of
canopy height improved the predicting power of the model. The two models we retained are of
the form of Eq. (2) and Eq. (3):
$AGB_{LCA} = a\,LCA + b$                                                          (2)
$AGB_{LCA} = (a\,LCA + b) \times WD$                                    (3)
where WD is the mean wood density of a site or the mean wood density of trees >50 cm in DBH
of a site.
We evaluated our results by applying a jackknife validation to our regression model, based on
1000 iterations of bootstrapping.  We also compared AGB as derived from LCA ($AGB_{LCA}$) to the
Lidar derived aboveground biomass ($AGB_{Lidar}$) in the nine $1km^2$ images. Coefficients of
correlation ($R^2$), root mean square error (RMSE) and bias are reported. We finally compared
these results to a traditional model relying on MCH to estimate AGB. The analysis was
performed using the R statistical software (R Core Team, 2014).

**2.5    Detecting Changes of Selectively Logging**
Forest degradation due to selective logging is difficult to detect with conventional remote
sensing techniques due to small scale and minor impacts on the forest canopy and biomass
compared to severe forest disturbances (e.g. fires, storms, or clearing). However, selective
logging targets large trees (Pearson et al., 2014) and thus may be detectable using LCA. Here, we
use the Antimary study site that was selectively logged after the 2010 Lidar acquisition to
examine the use of LCA for detecting logging impacts on the forest canopy and AGB.  We apply



the large tree segmentation approach on both the 2010 and on a 2011 post-logging Lidar data
(see Andersen et al., 2014 for details) to quantify the logging impacts in terms of the distribution
of large trees removed from the forest and the loss of aboveground biomass.

## 220   3      Results

### 221   3.1     Intersite comparison of landscapes and MCH

Topographic variation ranged from about 4 m elevation gain in flat area of Tapajós to steep
elevation gain of up to about 100 m in Cotriguaçu and Chocó (Fig. S3). Top canopy height
reached up to 60m, but varies across sites, with Chocó having the lowest MCH (24.1 m) and
Nouragues the highest (29.7 m). Forest height in Manaus was more homogeneous than in the
other sites, with a standard deviation of 6.8 m for MCH, versus 10.3 m in Paracou. We found no
relationship between topography and canopy height, which suggests that variability in forest
structure may be due to other ecological and edaphic factors in each site.
### 231   3.2     Large canopy area index

The choice of the canopy height threshold impacted LCA more than the minimum number of
pixels per cluster (Table S2). The difference due to the choice of the minimal cluster size
threshold was on average 1.4 %, calculated as the mean of the difference between the smallest
grain (50 pixels) and the largest one (200 pixels) across sites and height thresholds. Based on this
analysis, we chose to define LCA using a minimum cluster size of 100 pixels (100 m$^2$ for crown
area) in the remainder of this study. This corresponds to an area of at least 10 m x10 m or a circle
of approximately 11m in diameter, consistent with the average crown diameter of large trees of
the region (Bohlman and O'Brien, 2006; Figueiredo et al., 2016; Clark, unpublished results).




In contrast, the canopy height thresholds markedly impacted the magnitude of LCA among sites
(Fig. 1 and Fig. 2, Table S2). As the height threshold increased, intra-site variation of LCA(h)
became apparent, showing differences of LCA associated with differences of forest structure
(Fig. 1). Tapajós and Nouragues stood out with more area of large trees at the height threshold of
30 m ($LCA_{30m}$ = 51 and 48 %, respectively) , while Antimary and Chocó showed much lower
LCA at this height threshold ($LCA_{30m}$ = 21 %) (Table S2). The steepest slopes of the LCA(h)
function corresponded to the highest sensitivity of LCA to height thresholds and the inflection in
LCA was found between 24m in Antimary and 30m in Nouragues (Fig. 2).  The average height
of the steepest slope was about 27 m, a value that was used as the optimal threshold across all
sites.
Regressing $AGB_{Lidar}$ and LCA showed that the highest coefficients of correlation between the
two metrics occurred between 23 m (Chocó) and 30 m (Tapajós) height thresholds (Fig. 3a),
explaining more than 75 % of AGB variation in each site. The same analysis repeated using
$AGB_{inv}$ and LCA at the calibration sites (Fig. 3b) also confirmed the earlier results showing the
best relationships corresponded to height thresholds are found to be between 27m (Nouragues
and Paracou) and 28m (BCI and La Selva), with maximum coefficients of correlation ranging
between 0.5 and 0.8.  Based on these results, we defined LCA as the cumulative area of  clusters
of the canopy height model greater than 27 m height and each more than 100 $m^2$.

**3.3    Variation of AGB derived from LCA**
$AGB_{inv}$ was found to depend linearly on LCA (Eq. 2), with a better coefficient of correlation and
RMSE than other models, such as a power law fit ($R^2_{linear}$ = 0.59, $RMSE_{linear}$ = 62.53  Mg ha$^{-1}$, vs.
$R^2_{power}$ = 0.54, $RMSE_{power}$ = 65.38). Although this model was unbiased (bias = 0.0 Mg, $bias_{cross\_val}$



= 0.16 Mg), there were clear differences among study sites (Fig. 4, Table 3). These differences
were largely explained by landscape scale differences in wood density, an important factor
representing the influence of species composition on the spatial variation of AGB. To explore the
contribution of wood density across the study sites, we computed the average wood volume as
the ratio of AGB divided by the average wood density (Fig. 4b).   The linear relationship
between LCA and wood volume yielded an estimate of the average total volume of forests
independently of the site characteristics, through Vol = a LCA + b (Table 3).
For AGB estimation, the model based on LCA weighted by WD gives the best result by bringing
$R^2$ up to 0.78 and RMSE down to 46.02  Mg ha$^{-1}$ (Fig. 4b, Fig. 5, Table 3, Eq. (3)), with $AGB_{inv}$
and $AGB_{LCA}$ falling around a one-to-one line in Fig. 5a. At all sites, RMSE values are between
20.87 and 42.22 Mg, except Nouragues, where RMSE remains large (71.21 Mg) due to high
biomass and several outliers from the linear relation.
Finally, we applied the model from Eq. (3) to all 1km$^2$ areas and compared the derived $AGB_{LCA}$
to $AGB_{Lidar}$ (see Sect. 2.2), for which local models based on MCH were used (Fig. 5b). Global
RMSE was found to be 34.72 Mg and RMSE per site varied between 20.79 Mg at BCI and 49.58
Mg at Manaus. Our ground calibrated LCA model defined by Eq. (3) had a similar performance
as the MCH based AGB model ($R^2_{MCH}$ = 0.79, $RMSE_{MCH}$ = 44.2 Mg, Table S3). These findings
show that relying on a fraction of the Lidar information gives comparable results as using
metrics depending on information from all pixels, such as MCH, highlighting the importance of
large canopy trees to estimate biomass. The relationship between LCA and other metrics derived
from ground data, such as Lorey's height or basal area, are presented in Table S4.

**3.4 AGB changes from logging**



The impacts of logging on the distribution of large trees and changes of AGB was detected by
simply deriving the LCA index from pre and post-logging Lidar data acquired in 2010 and 2011
respectively in Antimary (Fig. 6).  Difference in LCA between the two dates (2010–2011) (Fig.
6a) at 1 ha grid cell captured the areas of largest changes in the few months following logging
(logging took place between June and November 2011, Lidar data were collected in late
November 2011). The LCA approach was able to detect approximately a 17 % decrease in LCA,
from a mean LCA of 34.8 % in 2010 to 29.2 % in 2011.
The changes were also captured in the frequency distribution of large canopy trees before and
after logging (Fig. 6b) and the differences in the spatial distribution (Fig. 6c and 6d).
These changes in LCA correspond to a biomass loss of 15.2 Mg ha$^{-1}$ when integrated in equation
(2) and were of the same magnitude of the planned selectively logging removal rate (12–18 Mg
ha$^{-1}$ or 10–15 m$^3$ ha$^{-1}$ of timber volume) (Andersen et al., 2014).  Difference in the Lidar index
($\Delta LCA$) at the native resolution of 1 m (Fig. 6e) was able to capture both the location of all large
trees removed from the forest stand and partial regeneration and gap filling that occurred in the
forest between the two dates.

**4     Discussion**
**4.1 Inter-site Comparisons**
Cross-site studies on the structure of tropical forests have led to significant advances in our
understanding of tropical forest ecology (Gentry 1993; Phillips et al., 1998; ter Steege et al.,
2006). They have also yielded important insights on new techniques to predict carbon stocks
across regions (eg. Asner and Mascaro, 2014).  Comparison of sites in terms of MCH derived
for the study sites confirms that there is a strong regional variations of AGB with respect to



canopy height, and that East Amazonian sites tend to have much taller trees than Central and
Western Amazonia sites. This was already apparent in the canopy height maps produced by the
GLAS sensor (Lefsky, 2010; Saatchi et al., 2011; Simard et al., 2011). Comparing sites in terms
of LCA showed a similar pattern of larger trees, being relatively more present in eastern
Amazonia, notably in the French Guiana sites and Tapajos.   Our most southwestern site was
Antimary, in the state of Acre (Brazilian Amazon) and does not represent areas in the Peruvian
Amazon and western Amazon-Andes gradients. The site in Chocó is also unique in its
characteristics because of extremely wet condition and unknown disturbance history (e.g.,
selective logging). Additional lidar and ground measurements would be needed in western
Amazonia to further validate the patterns observed in this study.

**4.2    Physical Interpretation of LCA**
In this study, we introduced a simple structural metric that captures the proportion of area
covered by large trees over the landscape ( > 1 ha) and explained the variation in average forest
volume and biomass when weighted by wood density in nine sites of old growth Neotropical
forests. LCA cannot separate the crown areas of individual trees.  However, it is adapted for
large scale monitoring of forest volume and biomass change, as it is a robust and readily
accessible metric. For individual tree separation, complex and more computationally intensive
approaches are available (Ferraz et al., 2016).
In estimating LCA from Lidar data, we examined the spatial clustering properties of LCA and
found that the minimum cluster size was less important than the threshold of canopy height, as
long as the analysis focused on the relative covered area instead of on the density of large trees.
We found that using the percentage of the area covered by large canopy trees is an efficient way



of overcoming the problem of individual crown segmentation in Lidar data. LCA is related to
how trees reaching the forest canopy (above a certain height) fill the space and how this
characteristic may follow a spatially invariant scaling across tropical forests (West et al., 2009).

**4.3    Correlation between LCA and AGB**
The distribution of $R^2$ between LCA and AGB for (Fig. 3) is such that the maximum difference
in $R^2$ between a threshold of 25m and 30m is approximately 0.1, a negligible value. Hence, AGB
retrieval by LCA is relatively insensitive to the height threshold.  For most sites, except
Antimary, we found a height threshold such that LCA explains about 80–90 % of the variation of
AGB or total volume of the forests for each site (60–70 % when compared with ground plots).
Using a height threshold of 27 m for all sites reduced the $R^2$ by 0.04 on average (max = 0.08)
compared to the optimal height threshold for each site. Hence, the difference between the $R^2$ of
Lidar and ground plots is due to the relative correlation between MCH used in Lidar derived
biomass and LCA. Differences in Lidar characteristics for each site and differences in timing of
Lidar observations and ground plots further amplify this problem. Finally, a limit to how much
LCA can explain variation in AGB relates to forest structure and the AGB of small trees.
Potential differences in MCH among sites are due to footprint size, scan angle and return density
(Disney et al., 2010; Hirata, 2004; Hopkinson, 2007). However, these effects are generally
smaller than the 1m increment that we used to determine the optimal height thresholds of LCA.
As a result, LCA estimation, and therefore AGB inferred from LCA, should depend little on
instrument, acquisition and processing (Table 2).  This is an important finding given the
increasing variety of airborne Lidar sensors, and also given the pre and post-processing methods
available for monitoring tropical forest structure and aboveground biomass.  However,





determining whether the 27m threshold holds for LCA calculation across in the tropics would
require a validation at more study studies across continents.

**4.4      LCA Relation to Ground Measurements**
The relation between LCA derived from Lidar and the ground measurements can be investigated
by converting the 27 m height threshold into equivalent DBH values, using a height–diameter
relationship.  In the absence of a local DBH–height relation at each site, we made use of the
following equation (Chave et al., 2014):
$$ln(H) = 0.893 - E + 0.760 \times ln(D) - 0.0340 \times (ln(D))^2 \qquad (4)$$
where E is a measure of environmental stress for each site that potentially impacts the tree
allometry. The corresponding DBH values fall around 35–55 cm, except for Chocó, where the
best coefficient of correlation is reached with a DBH threshold of 29 cm (Fig. S4). The DBH
estimation suggests that using a minimal DBH threshold of about 50 cm for large trees for old
growth neo-tropical forests better represents the total AGB variations.
The lower range of biomass estimation for the LCA model, associated with the intercept for LCA
equal to zero, ranged between 122  Mg ha$^{-1}$ in La Selva and 192  Mg ha$^{-1}$ in Paracou (Fig. 7a).
This lower range identified with the intercept of the LCA–AGB linear model can be interpreted
as the AGB associated with all trees smaller than 27 m and representing the smaller trees
(approximately all trees with DBH <50 cm). Note that the differences between sites are only due
to differences in their mean wood density and not the volume of trees (see Eq.(3) and Fig. 4).
Similarly, the contribution of small trees to the total biomass in the ground inventory ranges
between around 100 and 200  Mg ha$^{-1}$, except in Paracou (261  Mg ha$^{-1}$) (Fig. 7b).  AGB
estimation based on LCA in these sites cannot go under 100  Mg ha$^{-1}$ or over 500  Mg ha$^{-1}$. This



is not a limitation of the model because LCA is designed to provide AGB estimates for forests
reaching at least 27 m in mean canopy height, and such forests generally exceed 100 Mg ha$^{-1}$ in
AGB. Also, the upper threshold of 500 Mg ha$^{-1}$ is consistent with upper values found globally at
1 ha scale (Brienen et al., 2015; Slik et al., 2013). A recalibration of the method should be
envisaged in secondary and highly degraded forests.

**4.5    LCA as AGB Estimator**
The correlation of LCA to AGB$_{inv}$ suggests that a Lidar based approach can lead to the
estimation of AGB at the landscape scale and give useful information on the presence of large
canopy trees and their distribution, extending the analysis of large trees in plot level inventory
based studies (Bastin et al., 2015; Slik et al., 2013).
Therefore, LCA can explain the variations of total forest volume without any ancillary data
about the forest or the landscape. Any bias in conversion of LCA to AGB, however, can be
corrected across landscapes and sites by scaling the LCA–AGB relationship with average wood
density at the landscape scale.
Wood density has been shown to be a key element of allometric models of AGB estimation
(Baker et al., 2004; Brown et al., 1989; Chave et al., 2004; Nogueira et al., 2007). If wood
density is assumed to be constant across DBH classes, the mean wood density at the plot scale
can readily be used to scale LCA to biomass. However, if the wood density of large trees is
smaller or larger than the average wood density, (e.g. in BCI and Chocó: S.3, Fig. S5), the use of
mean wood density to scale LCA may introduce a slight bias in biomass estimation. A difference
in mean wood density of 0.1 g cm$^{-3}$ would introduce a bias of ±10 % in the biomass estimation
when using our model. We found that using mean wood density of large trees or basal area

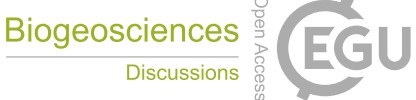

weighted wood density instead can give slightly better results and could circumvent the
differences in size distribution of the wood density (S.3).  Instead we could rely on the wood
density of large trees only. This would make the collection of ground data easier and cost
effective for biomass estimation, because trees ≥50 cm DBH only represent 5–10 % of the stems
of a plot (S.3, Fig. S6). Focusing on the wood density of dominant or hyper dominant species
could also be an alternative approach for future use of Lidar derived LCA for large scale biomass
estimation (Fauset et al., 2015; ter Steege et al., 2013).

Both MCH and LCA–AGB models performed relatively poorly in high biomass plots of the
Nouragues study area, by underestimating biomass values >500  Mg ha$^{-1}$ (Fig. 4 and 5).  To
explain the underestimation, we performed three tests: 1. We examined the differences in the
ground estimated biomass values with and without tree height and found no significant impact in
reducing the effect of underestimation. 2.  We tested the hypothesis that the height threshold
used for LCA estimation across sites was not suitable for the Nouragues study site and dismissed
the hypothesis because 27 m was found to be the optimum threshold for Nouragues plots. 3. We
examined the errors in the Lidar estimation of forest height and found that except for an
extremely high AGB$_{inv}$ of 617  Mg ha$^{-1}$, the four other high biomass outliers are all located in the
6 ha Pararé plot located on a very steep topography. The Lidar digital terrain model (DTM) of
this area shows an average within plots elevation range of 90 m. Ground detection on steep
terrain can be erroneous, depending on the Lidar point density and the view angle, causing large
area interpolation errors for DTM development and significant error in canopy height
measurements (Leitold et al., 2015). Other factors that may affect the underestimation of AGB



by LCA or MCH in the Nouragues site may be due to the presence of forest patches with clusters
of large trees and overlapping crown areas.

**4.6    LCA and forest degradation**
Although LCA and MCH may perform similarly in capturing the forest biomass variations and
changes, the use of LCA in detecting forest degradation and logging is more straightforward
because of its relation to large trees.    The LCA approach was able to accurately detect changes
in forests after logging by locating where the large trees are extracted.  Our estimate of biomass
change from  the LCA approach was higher than the biomass loss of 9.1  Mg ha$^{-1}$ reported by
another study using the 25$^{th}$ percentile height above ground as the Lidar metric for biomass
estimation (Andersen et al. 2014).  It can be expected that relying on the 25$^{th}$ percentile height
metric for biomass estimation would place more emphasis on the lower part of the canopy
(understory) that is either less damaged or has gone through some level of regeneration after
logging. Models based on LCA or MCH, on the other hand, may be more realistic for estimating
AGB changes because they capture the changes in large trees and upper forest canopy structure
that contain most of the biomass and are directly impacted by logging and biomass removal.

**4.7    Future Applications of LCA**
LCA definition in our study relies on the high resolution information on forest height, allowing
for the detection of crown area of large canopy trees.  Can a similar measure be derived from
large footprint Lidar observations such as the future NASA spaceborne Lidar mission GEDI
(Global Ecosystem Dynamic Investigation)?   GEDI will not provide spatially continuous data



on forest height, but its footprint size (~ 25 m) and dense sampling may be adequate to develop
statistical indicators of large trees over the landscape.
Similarly, future spaceborne radar missions could also provide useful information to retrieve
large canopy areas. The synthetic aperture radar (SAR) tomographical observations of the
European Space Agency (ESA) BIOMASS mission will provide wall-to-wall imagery of canopy
profile that could be converted to LCA over the landscape (Le Toan et al., 2011).  Preliminary
research based on airborne TomoSAR measurements has already shown that backscatter power
at about 30 m above the ground, with sensitivity to the distribution of large trees, explained the
variation of AGB over Nouragues and Paracou plots better than the backscatter power related to
the lower part of the canopy (0–15 m) (Minh et al., 2016; Rocca et al., 2014). Future research on
exploring the use of an equivalent radar index product from BIOMASS height or tomography
measurements at a height threshold (e.g. 27 m) may provide a potential algorithm to map the area
of large trees and estimate forest volume and biomass changes across the landscape.

**5      Conclusions**
We introduce LCA as a new Lidar derived index to capture the variations of large trees and total
volume and biomass across landscapes that remain spatially and regionally invariant.  The
importance of LCA is in its relevance to the structure and ecological characteristics of large trees
in filling the canopy space and their unique contribution in determining the total volume and
biomass of forests.  Unlike other Lidar derived metrics, LCA is linearly related to total
aboveground biomass after being weighted by average wood density and this linear relationship
remains unique across different forest types.  The comparison of LCA index with ground plots
suggests that DBH >50 cm is a more reliable threshold to quantify the number and distribution of




large trees and in capturing the variations of the total aboveground biomass across landscapes
and regions.

**Author contribution**
V. Meyer and S. Saatchi developed the model and designed the study. V. Meyer developed the
model code and performed the analysis. J. Chave, G. Vincent, M. Keller, F. Espírito-Santo, D.
Clark and M. d'Oliveira provided inventory data and derived metrics necessary to run the
experiments. A. Ferraz contributed to the data processing. D. Kaki performed a preliminary
analysis of the data. V. Meyer prepared the manuscript with contributions from all co-authors.

The authors declare that they have no conflict of interest.

**Acknowledgements**
The work described in this paper was carried out at the Jet Propulsion Laboratory, California
Institute of Technology, under contract with the National Aeronautics and Space Administration.
This work has benefited from "*Investissement d'Avenir*" grants managed by the French *Agence*
*Nationale de la Recherche* (CEBA, ref. ANR-10-LABX-25-01 and TULIP, ref. ANR-10-LABX-
0041; ANAEE-France: ANR-11-INBS-0001) and from CNES (TOSCA project; PI T Le Toan).
Field and Lidar data from the Brazilian sites were acquired by the Sustainable Landscapes Brazil
project supported by the Brazilian Agricultural Research Corporation (EMBRAPA), the US
Forest Service, and USAID, and the US Department of State. La Selva field work was supported
by the U.S. National Science Foundation LTREB Program NSF LTREB 1357177. Data in Chocó
are available as part of the Reducing Emissions from Deforestation and forest Degradation
(REDD) project. FES was supported by Natural Environment Research Council (NERC) grants
('BIO-RED' NE/N012542/1 and 'AFIRE' NE/P004512/1) and Newton Fund ('The UK
Academies/FAPESP Proc. N°: 2015/50392-8 Fellowship and Research Mobility'). The AGB
data for Paracou were made available courtesy of CIRAD (B. Hérault).




**Data accessibility**

The BCI lidar and forest inventory dataset used in this research are publically available from the
Office of Bioinformatics, Smithsonian Tropical Research Institute. All relevant data are within
the paper and its Supporting Information files.

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






**Table 1.** Information on forest inventory plots. * indicates that a site has been used for the calibration of the LCA
model. Sources: Antimary and Cotriguaçu: Fearnside, 1997; d'Oliveira et al., 2012, BCI: Center for Tropical Forest
Science (CTFS) (Condit, 1998; Hubbell et al., 1999, 2005), Chocó: (bioredd.org), La Selva: Carbono project (Clark
and Clark, 2000), Manaus and Tapajós: Espírito-Santo (unpublished results), Nouragues: Réjou-Méchain et al.,
2015, Paracou: Gourlet-Fleury et al., 2004; Vincent et al., 2012.

| Site | Data | Plots Size (ha) | N plots | Year | Mean WD (g cm$^{-3}$) | Mean AGB (Mg ha$^{-1}$) | Annual rainfall (mm yr$^{-1}$) |
|---|---|---|---|---|---|---|---|
| **Antimary (Brazil)** | Plot level | 0.25 | 50 | 2010 | 0.61 | 234 | 2000 |
| **BCI \* (Panama)** | Tree level | 1 | 50 | 2010 | 0.56 | 235 | 2600 |
| **Chocó (Colombia)** | Tree level | 0.25 | 42 | 2013 | 0.60 | 224 | 10000 |
| **Cotriguaçu (Brazil)** | Not available | - | - | - | 0.60 | - | 2000 |
| **La Selva \* (Costa Rica)** | Tree level | 1 | 11 | 2009 | 0.45 | 178 | 4000 |
| **Manaus (Brazil)** | Tree level | 0.25 | 10 | 2014 | 0.66 | 263 | 2200 |
| **Nouragues \* (French Guiana)** | Plot level<br>Tree level | 1<br>1 | 33<br>7/33 | 2012 | 0.66 | 424 | 3000 |
| **Paracou \* (French Guiana)** | Plot level | 1 | 85 | 2009-10 | 0.71 | 353 | 3000 |
| **Tapajós (Brazil)** | Tree level | 0.25 | 10 | 2014 | 0.62 | 238 | 1900 |





**Table 2**. Information on Lidar data and locations of the 9 research sites.

| Site (1km² images) | Sensor | Year | Returns m⁻² | Flight Altitude (m) | Scanning angle (°) | Frequency (kHz) | NW corner lat | NW corner lon |
|---|---|---|---|---|---|---|---|---|
| Antimary | Optech ALTM3100EA | 2010-2011 | 10-15 | 500 | 11 | 70 | 9°17'47.26"S | 68°17'15.06"W |
| BCI | Optech ALTM3100EA | 2009 | 8 | 1000 | 35 | 70 | 9°9'28.56"N | 79°51'18.9"W |
| Chocó | Optech ALTM3033 | 2013 | 4 | 1000 | 20 | 33 | 3°57'5.71"N | 76°49'10.31"W |
| Cotriguaçu | Optech ALTM3100EA | 2011 | 10-15 | 850 | 11 | 60 | 9°27'8.87"S | 58°51'51.22"W |
| La Selva | Optech ALTM3100EA | 2009 | 4 | 1500 | 20 | 70 | 10°25'37.97"N | 84°1'8.76"W |
| Manaus | Optech ALTM3100EA | 2012 | 10-15 | 850 (max) | 11 | 60 | 2°56'38.48"S | 59°56'12.57"W |
| Nouragues | Riegl LMS-Q560 | 2012 | 12 | 400 | 45 | 200 | 4°3'10.0"N | 52°42'19.95"W |
| Paracou | Riegl LMS-280i | 2009 | 4 | 120-220 | 30 | 24 | 5°15'47.73"N | 52°56'26.96"W |
| Tapajós | Optech ALTM3100EA | 2011 | 10-15 | 850 (max) | 11 | 60 | 2°50'53.41"S | 54°57'44.53"W |







**Table 3**. Coefficients, $R^2$, RMSE and bias for the models used to estimate $AGB_{LCA}$ without and with wood density
as a weighting factor (m_LCA) and m_LCA_wd, respectively).

| Model | Equation | a | b | $R^2$ | RMSE | Bias | $R^2$ cross-val | RMSE cross-val | Bias cross-val |
|---|---|---|---|---|---|---|---|---|---|
| **m_LCA** | AGB = aLCA + b (Eq. (2)) | 3.56 | 136.91 | 0.59 | 62.53 | 0.0 | 0.58 | 63.26 | 0.16 |
| **m_LCA_wd** | AGB = (aLCA+b) × WD (Eq. (3)) | 4.47 | 270.27 | 0.78 | 46.02 | -0.76 | 0.77 | 46.47 | -0.63 |







**Figure 1**. Segmentation of the 1 km × 1 km images in each site using five canopy height thresholds. A minimum of 100 contiguous pixels was used as a segmentation threshold in all cases.

**Figure 2** : LCA in function of height thresholds in the nine study sites. The steepest slopes are between 24 m (Antimary) and 30 m (Nouragues), with an average of 27 m across sites. Steepness of slope was obtained by calculating the derivative of the sigmoid models charactering each site.

**Figure 3**. Distribution of $R^2$ between tree height thresholds used to determine LCA and $AGB_{Lidar}$ in the nine 1 ha subareas (a) and distribution of $R^2$ between tree height thresholds and $AGB_{inv}$ in 1 ha inventory plots of the four calibration sites (b). All optimal thresholds are between 23 m and 30 m. The average maximal height threshold is 27 m.

**Figure 4**. Relationship between $AGB_{inv}$ density and LCA (a) and AGB density normalized by averaged wood (b). Normalizing AGB by averaged wood density brings the data from different sites closer to a common fit.

**Figure 5**. $AGB_{inv}$ density vs. $AGB_{LCA}$ estimated with LCA_wd model (a). $AGB_{Lidar}$ density from the 1km$^2$ images vs. $AGB_{LCA}$ estimated with LCA_wd model (b). The black line represents the 1-to-1 line.

**Figure 6**. Detection of changes of forest structure from selective logging in the Antimary study area showing a) the difference between pre- and post- logging (2010–2011) Lidar derived LCA at 1 ha grid cells over the entire study area, b) the histogram of LCA for the two Lidar datasets showing the mean difference and the reduction of medium and large LCA areas from selective logging, c) 2010 Lidar LCA segmentation at 1 m resolution over a sample area in the north of the study site, d) same LCA segmentation for 2011 Lidar data, and e) difference of the two segmented areas showing the extent of the logging impact on large trees in addition to natural changes of forest structure from changes in canopy gaps from tree falls and tree growth.

**Figure 7**. Relationship between LCA and $AGB_{LCA}$ (a) and relationship between $AGB_{inv}$ of large trees (>50 cm DBH) and total $AGB_{inv}$ (b). In both cases, the intercepts represent the contribution of small trees to total AGB. Note that Manaus and Nouragues overlap because they have the same mean wood density, as well as Chocó and Cotriguaçu.

813
814




815    Figure 1

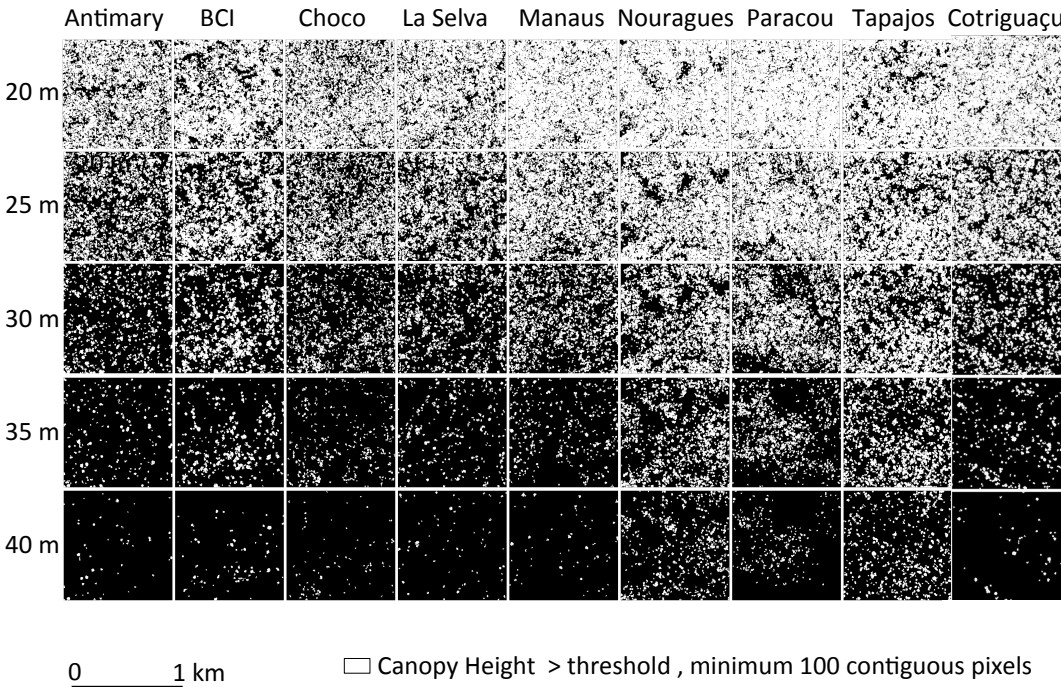

0 ——————— 1 km          ☐ Canopy Height > threshold , minimum 100 contiguous pixels







Figure 2

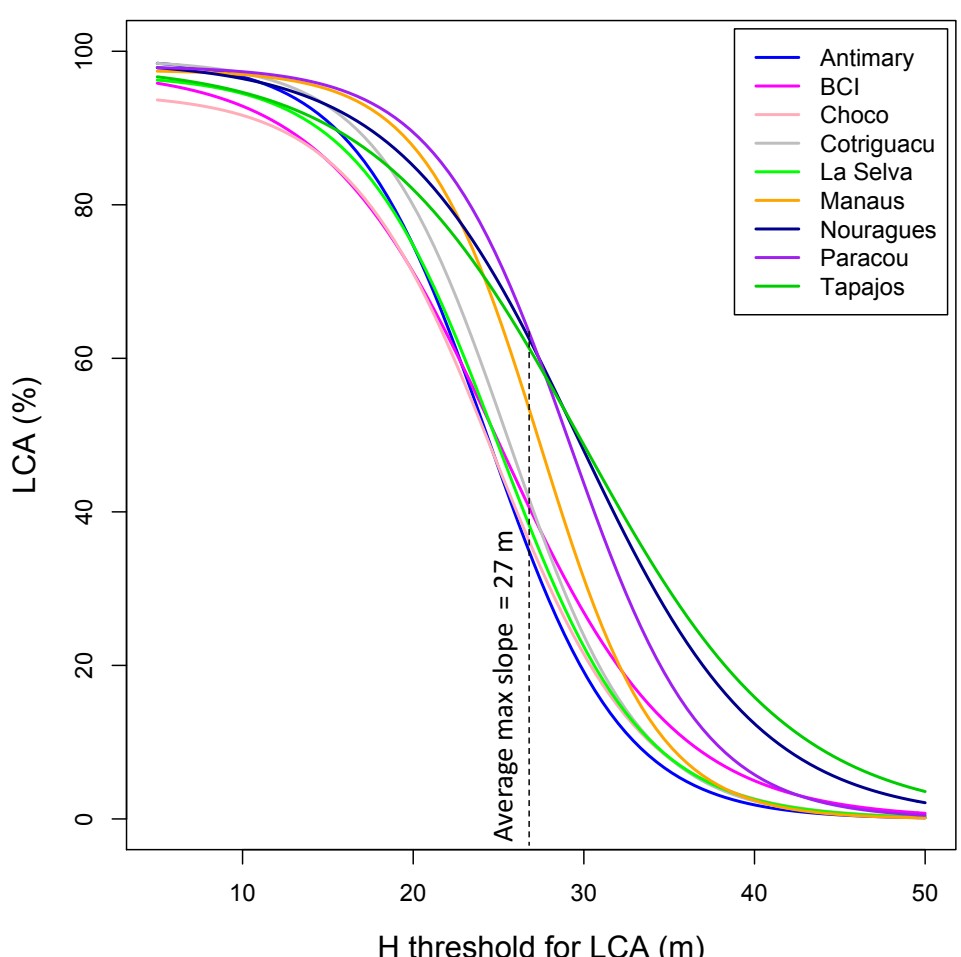





Figure 3

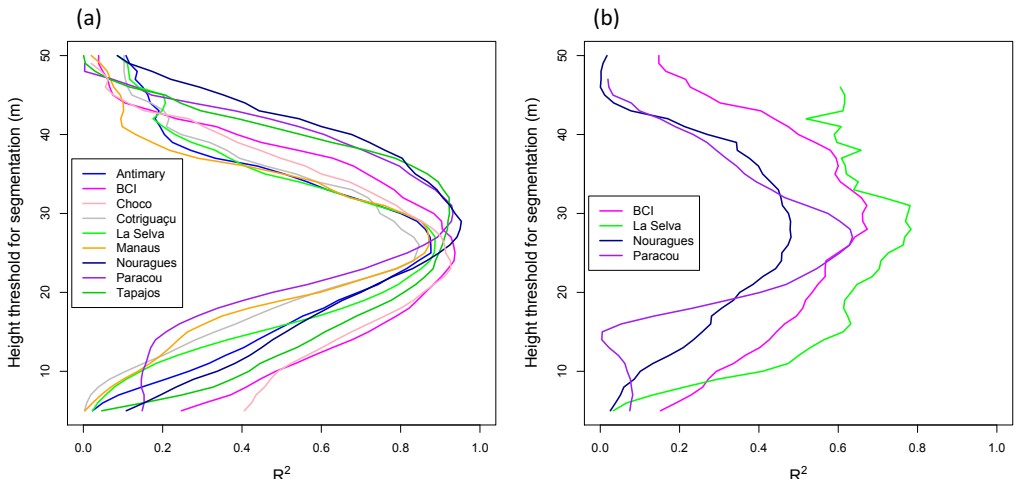







Figure 4

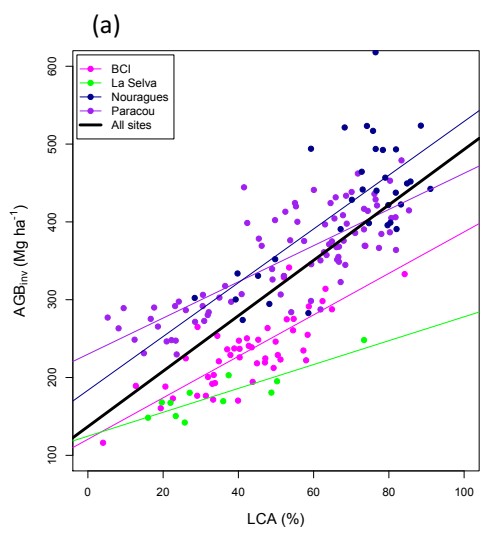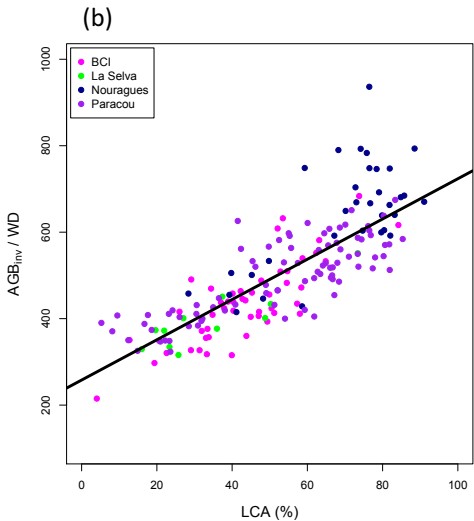





Figure 5

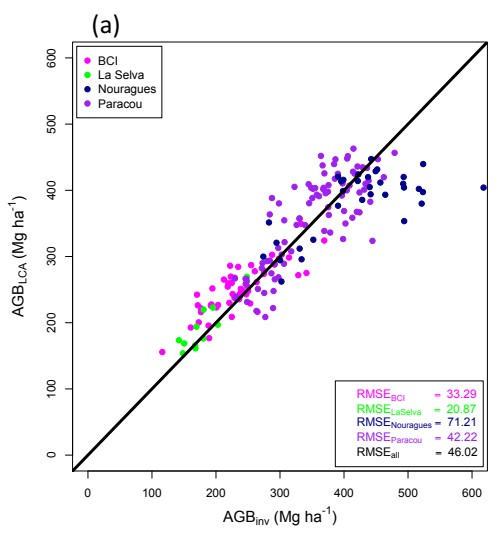
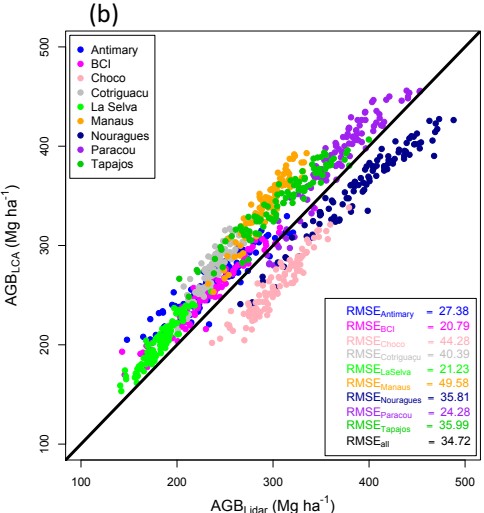






Figure 6

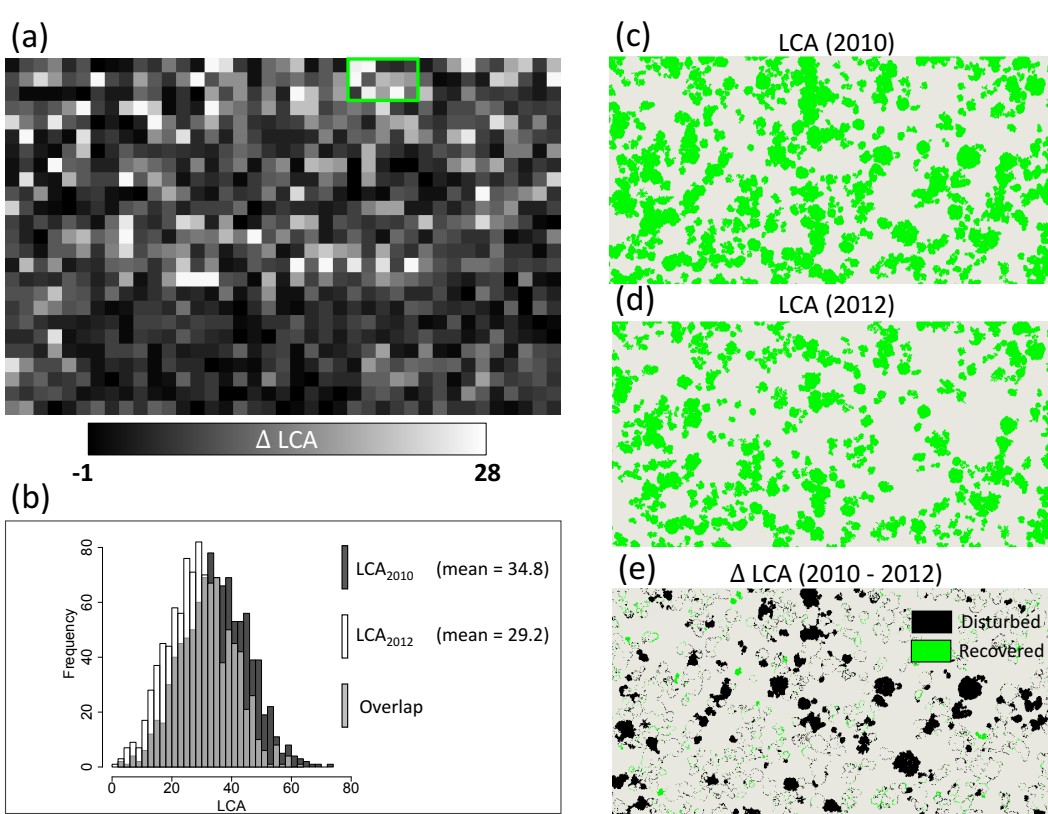







Figure 7

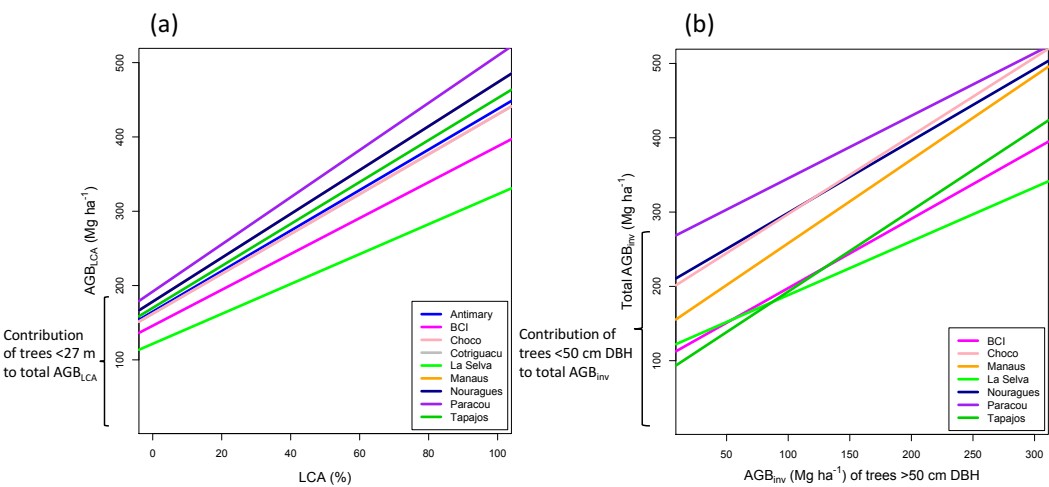
