# Peer review of "Canopy Area of Large Trees Explains Aboveground"

_Biogeosciences, 2017_

## Referee Comment (RC1) · Anonymous Referee #1 · 28 Jan 2018

General comments

This paper addresses an important and interested topic – the use of emergent crown area (from airborne LiDAR data) to estimate forest biomass. The abstract and introduction are very well written. Unfortunately, I get confused in the methods. From the abstract, it seems that the use of LCA to estimate biomass is going to be calibrated with ground AGB estimates. However, in sections 2.2 and 2.4 the authors estimate AGB from both LiDAR MCH and LCS. In the methods section, it is unclear whether they are predicting AGB_Lidar and AGB_LCA from an equation that already exists or whether they are doing a regression analysis to find values for parameters 'a' and 'b' in

[Figure]

Eqs. 1-3. If it's the former, show the actual values for 'a' and 'b'. Either way, it doesn't seem necessary to predict AGB from MCH other than to compare AGB estimates from LCA to those from MCH (eg, show improvement in new method). In section 2.3 the authors, say they have only 4 calibration sites (instead of 9 in the abstract). So, is AGB in the other five sights predicted by Eq 1 (MCH)?

This is important work, and I am happy to see it finally coming into fruition. However, I am strongly opposed to using estimates (ie, from MCH) to calibrate a new method. I suggest the authors remove AGB_Lidar estimates and focus on relating LCA metrics to AGB determined from ground inventories. This will also clean the paper - making it shorter and much easier to follow. Furthermore, I suggest trying to optimize AGB estimates from LiDAR by, for example, estimating AGB with both LCA and MCH. If I have misinterpreted the methods, please edit the manuscript accordingly. I look forward to seeing these improvements.

Specific comments and technical corrections

How is the LCA method weighted by WD if there isn't ground data at 5 sites? Line 104: what do you mean by 'unique'? Line 166: What model? Line 167: what data? Lines 203-4: This indicates that AGB_LCA is being tested against AGB_Lidar, where LiDAR is being treated as the reference. AGB_Lidar is only an estimate. Lines 205-6: Here you say that these results were compared to 'a traditional model relying on MCH to estimate AGB'. Isn't AGB_Lidar the model relying on MCH to estimate AGB? Section 2.5: Is it possible to apply the same methods to logged areas, since you may not know which areas have been harvested or not – or have before and after pictures? Line 269: Where did wood volume data come from? Lines 315-6: In what way does Antimary not represent Peruvian Amazon and Amazon-Andes gradients? Line 323: by how much does it explain the variation? Section 4.3: Would be helpful to refer to tables and figures Lines 344-6: This sentence is unclear to me, but it sounds like it supports my point that using AGB_Lidar as a reference is circular and not proving anything Line 374: Change 'only' to 'primarily' or something similar. Line 391: Change 'Any' to 'Most'

Lines 423-5: Maybe the relationship is not linear at the high end of LCA Line 467: If the relationship remains unique across forest types, is it not then broadly applicable? Fig 3: Clever way to find the optimal H threshold Fig 4b: This doesn't look like a perfectly fit Fig 5b: All calibration sites are above the 1:1 line. Why are Nouragues and Choco below the line? Fig 7: It would be helpful to see the actual data, not just regression lines.

---

## Referee Comment (RC2) · Anonymous Referee #2 · 21 Feb 2018

General comments: This paper presents an interesting new method for estimating aboveground biomass from Lidar data, the Large Tree Canopy Area metric, which is particularly interesting as it sits between area-based metrics and tree-centred approaches. For this method to be useful, it must either (1) outperform existing methods, (2) perform similarly to existing methods but at lower computational cost or (3) open up new applications not allowed by existing methods. The paper is framed around comparing the new LCA method against the existing MCH method, but a clear comparison of the two against ground-based validation data is not presented. From Table S3 it looks like MCH performs better in terms of RMSE and Rsq when wood density is not used (but with higher bias?), and LCA and MCH have similar performance when wood

density is used. Much of this paper compares biomass estimates from LCA and MCH, and finds that LCA is a good estimator of MCH. Is LCA quicker to calculate than MCH? It would be useful to present a comparison of the computational time taken to calculate LCA versus MCH. To me, the most compelling reason to use LCA is that it gives more information about forest structure than MCH. The application to detect the impacts of selective logging is potentially very important.

My main suggestion to improve this paper are to concentrate on testing the relative performance of LCA and MCH approaches at estimating biomass when validated against inventory data (even if LCA performs worse, this is still a very useful result for method development), and comparing the performance of the two approaches when applied to detect the impacts of selective logging. I agree with reviewer 1 in that I don't see much value in testing the performance of LCA against biomass estimates using MCH.

Specific comments: Line 205 – How was bias calculated? Line 262 – What are the other models apart from a power law fit? Line 262 – 263 – Are RMSE values and r squared values here from cross-validation or from the training data? Line 263 – Just present the bias from cross-validation. Line 271 – How feasible is it to scale by wood density in the absence of inventory data? Presumably errors would be larger if modelled estimates of wood density were used. Lines 287-301 – It would be useful to also see how MCH performs at detecting this loss of biomass. Lines 376-377 – This is a very nice approach to identify how much biomass is missed by LCA. Figure S2 - Given that the minimum cluster size didn't have a major effect on the AGB estimates, I would be interested in seeing a comparison of the performance of the LCA metric just following masking versus the LCA metric following removal of segments below the threshold cluster size. How computationally costly are these last steps?

Technical comments: Inconsistent approach to using capitals in section headings. Line 209 – => Detecting changes of selective logging. Line 385 - => LCA as an AGB estimator.

---

## Author Comment (AC2) · 31 Mar 2018

Thank you for taking the time to review our paper. We did our best to address all your comments in the hope this will improve the quality of the manuscript. Please note that all references to changes in manuscript correspond to the line numbers of the revised manuscript with track changes.

Comment: For this method to be useful, it must either (1) outperform existing methods, (2) perform similarly to existing methods but at lower computational cost or (3) open up new applications not allowed by existing methods.

Response : Our study does open up new applications compared with existing methods. We demonstrate that our method performs similarly to another method relying on information from all trees within a plot (MCH). The point of our paper is not to say that the LCA method is better than the MCH method, but rather to show that information on large trees is enough to estimate biomass. Our findings confirm what has been shown in several studies focusing on ground data (Bastin et al, Slik et al...) and shows for the first time that relying on large trees from a remote sensing perspective allows to estimate AGB. It opens up new applications both for field inventory and remote sensing applications. In the discussion (section 4.8), we talk about how methods focusing on large trees could help future space missions, such as BIOMASS and GEDI, to accurately estimate biomass and open up new applications. LCA also gives information on the presence of large trees in a study area, which other metrics such as MCH cannot do. It is an important point, considering that large trees are often the most affected by natural disturbance and targeted by logging companies. Changes to manuscript: ls.455-457: "LCA provides information on the presence of large trees in a study area, which other metrics such as MCH cannot do. It is an important point, considering that large trees are often the most affected by natural disturbance and targeted by logging companies." ls.564-565: "The comparison of LCA and MCH metrics showed that both performed similarly in estimating AGB, highlighting the importance of large canopy trees to estimate biomass." ls.645-647: "The results of our study may encourage further research in the use of Lidar data for detecting the distribution of larger trees in tropical forests for ecological and conservation studies."

Comment: The paper is framed around comparing the new LCA method against the existing MCH method, but a clear comparison of the two against ground-based validation data is not presented.

Response: Thank you for pointing this out. We added a short paragraph in the method section, as well as a new section in the Results and in the Discussion, comparing the performance of LCA and MCH methods. This is presented in the Methods (ls.218-240), in the Results (ls. 345-379) and in the Discussion (ls.563-569). To avoid any confusion, we moved the MCH local estimations of AGB from the main Lidar data paragraph to the Supplementary information (S.2). AGBLidar was also renamed LCALocal for clarity. Changes to manuscript: see ls. 218-240, ls. 345-379 and ls. 563-569. Figure 5 (attached here as Fig. 1) We chose to keep Table S3 in the Supplementary Information for clarity, but we added a figure comparing AGB estimations using the 2 methods (Figure 5, attached here as Fig.1).

Comment: Is LCA quicker to calculate than MCH? It would be useful to present a comparison of the computational time taken to calculate LCA versus MCH.

Response: LCA is not quicker to calculate than MCH, but it is not significantly slower either (below 1s for both methods). Also, the strength of LCA lies in the structural information it provides, not in its computational time. Thus, we chose not to add a detailed comparison of computational time.

Comment: The application to detect the impacts of selective logging is potentially very important.

Response: We agree. We emphasized this point in the Discussion: Changes to manuscript: ls.609-611: "LCA could become an important tool to detect forest degradation, in particular selective logging, considering that large trees are targeted by logging companies."

Comment: My main suggestion to improve this paper are to concentrate on testing the relative performance of LCA and MCH approaches at estimating biomass when validated against inventory data (even if LCA performs worse, this is still a very useful result for method development),

Response: Thank you for your suggestion. As mentioned above, we added a paragraph in the method section, as well as two new sections (results and discussion) and a figure comparing the two methods, showing that they perform very similarly. We also show how they differ in terms of AGB estimations in different sites.

Comment: and comparing the performance of the two approaches when applied to detect the impacts of selective logging.

Response: We compared the performance of the 2 approaches when applied to selective logging detection. The MCH model showed a loss of biomass of 19 Mg ha-1, compared to 15 with LCA and 9 from a previous study based on rh25. We added this information in the results and the discussion. Changes to manuscript: ls.393-394: "As a comparison, the MCH model led to an estimated biomass loss of 19 Mg ha-1." ls.607-609: "The higher biomass loss estimation from the MCH model (19 Mg ha-1) again shows how different metrics can lead to different results. Here, three methods based on three different Lidar metrics yielded results that differed by more than twofold.".

Comment: I agree with reviewer 1 in that I don't see much value in testing the performance of LCA against biomass estimates using MCH.

Response: Thank you for your suggestion. We removed Figure 5b. Performance comparison of LCA and MCH model at the calibration sites is now based on Figure 5a. The models applied to the nine sites are now Figure 5b, following your other suggestion to focus on the comparison of LCA and MCH methods.

Specific comments:

Comment: Line 205 – How was bias calculated?

Response: We added the definition of bias to the manuscript: Changes to manuscript: ls.214-215: "bias (mean difference between the expected values of AGB and the observed values of AGB)".

Comment: Line 262 – What are the other models apart from a power law fit?

Response: For both LCA and MCH models, we tested linear models and power laws, which are the 2 common fits. We modified the sentence to avoid any confusion:

Changes to manuscript: ls.302-303: "with a better coefficient of correlation and RMSE than a power law fit"

Comment: Line 262 – 263 – Are RMSE values and r squared values here from cross-validation or from the training data? Line 263 – Just present the bias from cross-validation.

Response: R2 and RMSE are from training data. We removed the bias from the training data and present the bias from cross-validation. Changes to manuscript: l.304: "biascross_val = 0.16 Mg" ls.334-336: "coefficients of correlation, RMSE and bias from training data and cross-validation are reported in Table 3."

Comment: Line 271 – How feasible is it to scale by wood density in the absence of inventory data? Presumably errors would be larger if modelled estimates of wood density were used.

Response: We agree. If there is no information in the literature from previous studies, modelled WD could be used, but would indeed give greater errors. This is now covered in the Discussion. Changes to manuscript: ls.558-561: "In the absence of information on wood density from the literature, modelled wood density could potentially be used, but would give greater errors. These errors should be taken into account when reporting on the uncertainty of the results."

Comment: Lines 287-301 – It would be useful to also see how MCH performs at detecting this loss of biomass.

Response: The MCH model (Table S3) gives a biomass loss of 19mg/ha, more than twice what was reported in Andersen et al., 2014. These results were added to the results section and the discussion section 4.6.: Changes to manuscript: ls.393-394: "As a comparison, the MCH model led to an estimated biomass loss of 19 Mg ha-1." ls.607-609: "The higher biomass loss estimation from the MCH model (19 Mg ha-1) again shows how different metrics can lead to different results. Here, three methods based on three different Lidar metrics yielded results that differed by more than twofold.".

Comment: Lines 376-377 – This is a very nice approach to identify how much biomass is missed by LCA.

Response: Thank you for this positive comment.

Comment: Figure S2 - Given that the minimum cluster size didn't have a major effect on the AGB estimates, I would be interested in seeing a comparison of the performance of the LCA metric just following masking versus the LCA metric following removal of segments below the threshold cluster size. How computationally costly are these last steps?

Response: This is a good point. For a reference image of 1000x1000m pixels, the full process takes less than one second. Just using masking may be slightly faster, but the computational cost is not an issue here. Just using masking gives similar results as when using LCA, because the pixels removed by the full process represent a small fraction of the area covered by large trees (1.73% on average). (R2=0.78, RMSE=45.7, bias=0.55) These isolated pixels either represent single branches reaching above 27m or the tip of a tree whose crown is mainly below 27m. Therefore, these pixels have no meaning in terms of our LCA metric and do not represent large trees. This is why we chose to remove them. The goal of our study is to show that large trees are sufficient to estimate AGB. We clarified this point in the manuscript: Changes to manuscript: ls.450-454: "Clusters smaller than 100 m2 add only a small fraction (1.7% on average) to LCA values across sites. Including these clusters in LCA would not impact the performance of the model (similar R2, RMSE and bias) and would allow to skip the final steps of the LCA retrieval (see Fig. S2). However, since these pixels either represent single branches reaching above 27m or the tip of a tree crown, they have no meaning in terms of our LCA metric and do not represent large trees.".

Comment: Technical comments: Inconsistent approach to using capitals in section headings. Line 209 – => Detecting changes of selective logging. Line 385 - => LCA as an AGB estimator

Response: Thank you for pointing this out. We removed the capital letters accordingly.

Please also note the supplement to this comment:
https://www.biogeosciences-discuss.net/bg-2017-547/bg-2017-547-AC2-supplement.pdf

[Figure]

**Fig. 1.** AGBMCH vs. AGBLCA in the plots of the four calibration sites (a), and AGBMCH vs. AGBLCA in the 1km2 images of the nine sites (b). The black line represents the 1-to-1 line.

**Supplement:**

**Supplement**

[Figure]

**Figure S1**. Location of the nine study sites (Globcover). All sites are located in old growth tropical forests.

[Figure]

**Figure S2** : Description of steps taken from the original Lidar canopy height model to the final LCA product. This
example is a 400 m by 400 m area in BCI, with a Height threshold of 30 m and the minimum number of contiguous
pixels set to 100.

[Figure]

Topography · Elevation · MCH · CHM

**Antimary**
to 204 m | 24.9 ± 7.5 m

**BCI**
to 178 m | 24.8 ± 8.9 m

**Chocó**
to 186 m | 24.1 ± 8.9 m

**Cotriguaçu**
to 371 m | 25.2 ± 7.2 m

**La Selva**
to 134 m | 24.8 ± 8.1 m

**Manaus**
to 104 m | 26.7 ± 6.8 m

**Nouragues**
to 151 m | 29.7 ± 9.5 m

**Paracou**
to 220 m | 28.9 ± 10.3 m

**Tapajos**
to 177 m | 29.1 ± 7.2 m

1 km m          60 m

**Figure S3**. Lidar derived images: shaded relief of topography and canopy height model (CHM), at 1 m resolution. Mean canopy height (MCH), standard deviation of canopy height and elevation range are reported.

**S.1 Estimating aboveground biomass from forest inventories.**

For trees with no height measurement, a site specific DBH height model was used to infer tree height in each site, as described in previous studies (Feldpauch et al., 2012; Meyer et al., 2013,). Wood density was extracted from the global wood density database for tropical trees (Chave et al., 2009; Zanne et al., 2009) for each tree, based on its level of botanical identification (species, genus, family). Trees with no botanical identification were assigned the average wood density of the plot. Average wood density of each site was calculated as the unweighted average of all trees within a site, or was taken from a previous study in the case of Cotriguaçu (Fearnside, 1997). Average wood density of large trees was calculated as the unweighted average of trees with DBH ≥50 cm in each site. Tree level AGB was aggregated at the plot level using a commonly used allometric regression model for moist tropical forests (Eq. (S1), Chave et al., 2005), except for La Selva and Chocó, for which a model for wet tropical forest (Eq. (S2), Chave et al., 2005) and a local allometric model (Eq. (S3), Duque et al., 2017) were used, respectively.

$$AGB_{\_moist} = 0.0509 \times \rho DBH^2 H \qquad \text{(S1)}$$

$$AGB_{\_wet} = 0.0776 \times (\rho DBH^2 H)^{0.940} \qquad \text{(S2)}$$

$$AGB_{\_Chocó} = 0.089 \times (\rho DBH^2 H)^{0.951} \qquad \text{(S3)}$$

where trunk diameter ($DBH$ in cm) is measured during the inventory, tree height ($H$, in m) is either measured in the field or estimated from a local DBH–H model, and specific gravity or wood density ($\rho$ in g cm$^{-3}$) is known for each tree. AGB (in kg of dry biomass) of individual trees estimated using the former equations was used to calculate plot level AGB density (Mg ha$^{-1}$) by summing over the biomass of all stems within each plot. Ground estimated AGB density is henceforth referred to as AGB$_{inv.}$ Estimating AGB with these allometric models has been reported to have a standard error of 12.5 % when using height as a parameter, against 19.5 %

when height is not available (Chave et al., 2005).

**S.2 Local estimates of AGB using MCH**

We estimated AGB locally for each of the nine sites in order to find the best height threshold for LCA in all sites, in addition to the information provided by the four calibration sites (see Fig. 3). Mean canopy height (MCH) is a good predictor of AGB provided that the regression model is calibrated locally. It was calculated by averaging all the canopy height model pixels falling in an area of interest. Here, we calculated a locally calibrated AGB map of each site from MCH using the following model form (Eq. (3), Asner and Mascaro, 2014).

$$AGB_{Local} = aMCH^b + \epsilon \tag{S4}$$

where $AGB_{Local}$ is the aboveground biomass estimation derived from Lidar data, $a$ is a scaling constant, which is expected to depend significantly on forest type and stand level wood density, $b$ is a power law exponent and $\epsilon \sim N(0, \sigma^2)$ represents the uncertainty in measurements. All coefficients are presented in Table S1. We inferred the model parameters directly for the sites where $AGB_{inv}$ of 1 ha plots was available (La Selva, BCI, Paracou and Nouragues). For Chocó and Antimary, we developed models based on 0.25 ha plots and 50 m x 50 m pixels of Lidar data and after estimating $AGB_{Local}$, aggregated the image to 1 ha or 100 m pixels. For the remaining sites of the Central Amazon (Cotriguaçu, Manaus and Tapajós), we developed a model based on existing data in Manaus and Tapajós from a previous study, derived from airborne and spaceborne Lidar (see 
[revised manuscript text omitted]

---

## Author Response (AR1)

To: Associate Editor, Jochen Schöngart

Dear Editor,

Thank you for handling this manuscript. We are pleased to see that the feedback from the reviewers was overall positive, and that they both suggested improvements to the manuscript, which we took onboard.

Both reviewers pointed out issues with the calibration of our new model (henceforth referred to as LCA model). Our methodology was not clearly stated. The LCA model was not calibrated from Lidar data but from ground data at 4 sites, and we edited the manuscript to avoid confusion about it. We developed the local models based on MCH to confirm the optimal height threshold for segmentation as indicated by Figure 2 and Figure 3. MCH-inferred AGB values are now just used as a test for validation of our height threshold in Figure 3. We also added sections comparing the LCA model to a similar model based on MCH calibrated from the same 4 sites, as suggested by the reviewers.

Comments made by both reviewers were addressed in the authors' comments as part of the interactive discussion process, and are presented here again, with additional information about changes that were made in the manuscript.

Please note that all references to changes in manuscript correspond to the line numbers of the revised manuscript with track changes.

We believe that these changes and the ones described below improved the clarity of our paper, and that it is now acceptable for publication in your journal.

Sincerely,

Victoria Meyer, on behalf of all co-authors.

**Response to Anonymous Referee #1**

**Response to General comments:**

Thank you for reviewing our manuscript. We greatly appreciate your comments and did our best
to address the issues you brought up. Your comments highlighted the fact that our methodology
was not clearly stated. The LCA model was calibrated using inventory data from the four sites
referred to as "calibration sites" in the manuscript. Based on both reviews of the paper, we
decided to remove Figure 5b and moved the paragraph explaining how $AGB_{Lidar}$ (renamed
$AGB_{Local}$ for clarity) was calculated to the Supplementary Information (S.2), to make the paper
more straight forward and focused on LCA. $AGB_{Local}$ values are now just used as a test for
validation of our height threshold in Figure 3.

**Comment**: "In the methods section, it is unclear whether they are predicting AGB_Lidar and
AGB_LCA from an equation that already exists or whether they are doing a regression analysis
to find values for parameters 'a' and 'b' in Eqs. 1-3. If it's the former, show the actual values for
'a' and 'b'."
**Response:** The form of Equation 1 (now Equation S4) is a commonly used model form to
estimate AGB from Lidar locally (see Asner and Mascaro, 2014). For each site (or group of sites
for Manaus, Tapajos and Cotriguaçu), we performed a regression based on that form and
obtained coefficients a and b, presented in Table S1 (SI, ls.50-51: "All coefficients are presented
in Table S1").
We decided to move this section to the Supplementary Information, as it is not central to the
paper and is just used to obtain Figure 3a in this new version of the paper.
Coefficients a and b for Equation 2) and 3) (now Eq 1 and 2) are presented in Table 3. We added
a sentence that makes a clear reference to the coefficients in that table. Also, we moved the
section presenting the form of the LCA models from the Methods to the Results section, for
clarity (ls.358-364).
**Changes to manuscript:** ls.334-335 "The coefficients of the models, as well as their respective
coefficients of correlation, RMSE and bias from all training data and cross-validation are
reported in Table 3."

**Comment**: "Either way, it doesn't seem necessary to predict AGB from MCH other than to
compare AGB estimates from LCA to those from MCH (eg, show improvement in new
method)."
**Response**: Based on both reviewers' comments, we removed the part of the analysis that
compared AGBLCA to the locally estimated $AGB_{Lidar}$. As a result, Figure 5b was removed.
Instead, we are now comparing AGB estimations from LCA and MCH based on the same
methodology: in both cases, models were fitted using the field $AGB_{inv}$ of the four calibration
plots. This is presented in the Methods (ls.218-240), in the Results (ls. 345-379) and in the
Discussion (ls.563-569).
**Changes to manuscript**: see ls. 218-240, ls. 345-379 and ls. 563-569. Figure 5

**Comment:** "In section 2.3 the authors say they have only 4 calibration sites (instead of 9 in the
abstract)."
**Response**: We realize that the abstract was misleading. We added a sentence stating that the
model was calibrated using 4 sites. We also removed the word "nine" in the title of the paper.
Changes to manuscript: ls.45-46: "…and ground inventory data in nine undisturbed old growth
Neotropical forests, of which four had plots large enough (1ha) to calibrate our model."

**Comment**: "So, is AGB in the other five sites predicted by Eq 1 (MCH)?"
**Response:** $AGB_{LCA}$ in the other sites was estimated using the same LCA model calibrated from
the 4 calibration sites (Eq 2). $AGB_{MCH}$ was calculated using the MCH model presented in Table
S3.

**Comment**: "I suggest the authors remove AGB_Lidar estimates and focus on relating LCA
metrics to AGB determined from ground inventories."
**Response**: Thank you for your suggestion. We removed figure 5b and removed the paragraphs
related to $AGB_{Lidar}$ in Section 2.2. The information on $AGB_{Lidar}$ (renamed as $AGB_{Local}$) are now
provided in the Supplementary Information (S.2). $AGB_{Local}$ is now only used to provide
additional information on the choice of the height threshold in Figure 3. (nb: equation numbers
have changed).
We edited the text to emphasize the role of the calibration plots and show that $AGB_{Local}$ was just
used as an additional/confirmation step.
**Changes to manuscript**: ls.200-206: "We determined the optimal minimum canopy height
threshold calculating the coefficient of correlation between $AGB_{inv}$ and LCA at the four
calibration sites.  (…).. We also estimated AGB from Lidar data locally ($AGB_{Local}$) using a
commonly used model fit relating MCH to $AGB_{inv}$ in each site, to further examine the variations
of LCA and AGB in all nine sites (see S.2, Table S1)."

**Comment:** "Furthermore, I suggest trying to optimize AGB estimates from LiDAR by, for
example, estimating AGB with both LCA and MCH."
**Response:** We tested different model forms for Equation 2 and 3 (now Equations 1 and 2),
including models using both LCA and MCH as predictors. Using MCH in addition to LCA did
not improve the performance of the model. This is stated in the sentence ls.234-237 "We tested
different models to infer $AGB_{inv}$ from LCA, henceforth called $AGB_{LCA}$, at the four calibration
sites, and explored if adding more parameters, such as mean wood density of a site, mean wood
density of large trees (DBH ≥50 cm), mean canopy height or top percentiles of canopy height
improved the predicting power of the moded." We added:
**Changes to manuscript:** ls.311-331"Adding more parameters did not improve the performance
of the model, except when using WD as a normalizing factor. The two models we retained are
therefore of the form of Eq. (1) and Eq. (2)"

**Responses to specific comments:**
**Comment:** How is the LCA method weighted by WD if there isn't ground data at 5 sites?
**Response**: Ground data are available in all sites except Cotriguaçu, but plot size was too small to
be used in the LCA model calibration process. Howewer, wood density estimation does not
depend on plot size, and wood density information was used from all sites to obtain a site-
averaged wood density (see Table 1). A sentence was added to highlight this point:

**Changes to manuscript**: ls.138-145: "For this reason, all plots smaller than 1 ha were excluded
from the LCA analysis but were used in estimating average wood density for each site, which
does not depend on plot size. Stand averaged wood density was calculated based on the wood
density of all trees present in a site, determined using the commonly used global wood density
database, and is reported in Table 1 (Chave et al., 2009; Zanne et al., 2009). For Cotriguaçu, we
used stand averaged wood density given by Fearnside, (1997) for a region covering the site."
**Comment:** Line104: what do you mean by 'unique'?
**Response**: by "unique", we mean one model that would work across sites in the Neotropics.
**Changes to manuscript:** l.112: We modified the sentence accordingly to "single".
**Comment**: Line 166: What model? Line 167: what data?
**Response:** The text was edited to clarify this sentence.
**Changes to manuscript:** SI, ls.55-58 "For the remaining sites of the Central Amazon
(Cotriguaçu, Manaus and Tapajós), we developed a model based on existing data in Manaus and
Tapajós from a previous study, derived from airborne and spaceborne Lidar (see Lefsky et al.,
2007)." Note that this section is now part of the Supplementary Information, as explained above.
**Comment**: Lines 203-4: This indicates that AGB_LCA is being tested against AGB_Lidar,
where LiDAR is being treated as the reference. AGB_Lidar is only an estimate.
**Response**: This is correct. The goal here was to test $AGB_{LCA}$ against locally derived $AGB_{Lidar}$.
Based on both reviewers' comments, we realized that this step was not necessary and was
removed from the paper.
**Changes to manuscript**: Figure 5b and any text related to this graph were removed from the
paper.
**Comment**: Lines 205-6: Here you say that these results were compared to 'a traditional model
relying on MCH to estimate AGB'. Isn't AGB_Lidar the model relying on MCH to estimate
AGB?
**Response**: Thank you for highlighting this point. Here, we refer to a single model based on
MCH from all the calibration sites, the same way that the LCA model was calibrated. This way,
we can compare the LCA model to a MCH model. We realize that this sentence is confusing and
edited the manuscript to clarify it: as stated above, $AGB_{Lidar}$ is now only used to obtain Figure 3b
and is no longer compared to $AGB_{LCA}$. Instead, we added a new section in the methods, results
and discussions comparing $AGB_{LCA}$ and $AGB_{MCH}$ (based on a model calibrated on the same 4
calibration sites). Please report to our response to earlier comment.
**Comment:** Section 2.5: Is it possible to apply the same methods to logged areas, since you may
not know which areas have been harvested or not – or have before and after pictures?
**Response**: We agree that we need before-after data to detect logging. In the example we are
showing, we do have before and after logging Lidar data. Details are provided in Anderson et al.,
2014.
We added a sentence to emphasize on the need for this type of datasets.
**Changes to manuscript**: ls.246-247: "provided that Lidar data are available from pre and post-
logging.".

**Comment:** Line 269: Where did wood volume data come from?

**Response**: we edited the manuscript to clarify this point:

Changes to manuscript: ls.307-309: "Since AGB depends on DBH, H and WD (see Chave et al.,

2014), average wood volume can be computed approximately as the ratio of AGB divided by the average wood density".

**Comment:** Lines 315-6: In what way does Antimary not represent Peruvian Amazon and

Amazon-Andes gradients?

**Response:** We added the following sentence to be more specific :

**Changes to manuscript:** ls.418-421: "However, this site does not represent forests in the western Amazon or the Amazon-Andes gradients with relatively lower wood density (Baker et al. 2004) and more fertile volcanic soils impacting the forest structure and dynamics (Quesada et al., 2011)."

**Comment:** Line 323: by how much does it explain the variation?

**Response:** Overall 78% is explained (R2=0.78).

**Changes to manuscript:** l.428: "and explained 78% of the variation".

**Comment**: Section 4.3: Would be helpful to refer to tables and figures

**Response**: Thank you for the suggestion. We added references to table 2 and figure 3.

Changes to manuscript:  references l.465 and l.468.

**Comment**: Lines 344-6: This sentence is unclear to me, but it sounds like it supports my point that using AGB_Lidar as a reference is circular and not proving anything

**Response**: This sentence was not clear and was removed from the manuscript. Moreover, we are now comparing AGB from LCA and MCH in a separate section of the results and discussion to avoid any confusion.

**Comment**: Line 374: Change 'only' to 'primarily' or something similar.

**Response**: "only" was removed.

**Commen**t: Line 391: Change 'Any' to 'Most'

**Response**: We changed "'Any' to 'Most'.

**Comment:** Lines 423-5: Maybe the relationship is not linear at the high end of LCA

**Response**: It is indeed a possibility. We added this suggestion to the manuscript.

Changes to manuscript: ls. 589-591: "It is also possible that the relationship between AGB and

LCA is not linear for very high AGB values. This could be tested in the future with a larger number of sites with very high biomass."

**Comment:** Line 467: If the relationship remains unique across forest types, is it not then broadly applicable?

**Response**: Yes, this is an important point of the paper. We added two sentences highlighting this fact.

**Changes to manuscript**:

- in the Discussion:

ls.538-539: "Our model can therefore potentially be applied to a wide range of forest types,
provided that there is information about wood density of the study area in the literature."
- in the Conclusion:
ls.640-641: ". This linear relationship remains unique across different forest types, making the
LCA model broadly applicable."
**Comment**: Fig 3: Clever way to find the optimal H threshold
**Response**: Thank you for this positive comment.
**Comment:** Fig 4b: This doesn't look like a perfectly fit.
**Response:** With a R2 of 0.78, RMSE of 46 and no bias, we consider the fit to be good. These
number are provided in Table 3. R2 was added to Figure 4b to emphasize this point.
**Changes to manuscript:** R2 was added to Figure 4b to emphasize this point.
**Comment:** Fig 5b: All calibration sites are above the 1:1 line. Why are Nouragues and Choco
below the line?
**Response:** Based on your comments and that of Reviewer 2, we removed this figure. The fact
that some plots were above/below the line was likely due to the fact that $AGB_{Lidar}$ was estimated
locally for different sites and included some error. We are now simply comparing the LCA and
MCH methods based on the inventory data only (Figure 5, attached here as Fig.1).
**Comment:** Fig 7: It would be helpful to see the actual data, not just regression lines.
Response: The point of this figure is to clearly see where the lines cross the y axis. For Fig 7a),
we are just showing where the LCA model crosses the y axis, with different wood density from
the different sites. Each line represents the model curve with various wood density values. To see
the actual data from the calibration sites, see Figure 4b.
For fig 7b, actual data could be added, but just showing the lines gives the figure a clean look,
considering that the information we are looking for here is the intercept of each line.

**Response to Anonymous Referee #2**

Thank you for taking the time to review our paper. We did our best to address all your comments
in the hope this will improve the quality of the manuscript.
**Comment:** For this method to be useful, it must either (1) outperform existing methods, (2)
perform similarly to existing methods but at lower computational cost or (3) open up new
applications not allowed by existing methods.
**Response :** Our study does open up new applications compared with existing methods. We
demonstrate that our method performs similarly to another method relying on information from
all trees within a plot (MCH). The point of our paper is not to say that the LCA method is better
than the MCH method, but rather to show that information on large trees is enough to estimate
biomass. Our findings confirm what has been shown in several studies focusing on ground data
(Bastin et al, Slik et al…) and shows for the first time that relying on large trees from a remote sensing perspective allows to estimate AGB. It opens up new applications both for field
inventory and remote sensing applications. In the discussion (section 4.8), we talk about how
methods focusing on large trees could help future space missions, such as BIOMASS and GEDI,
to accurately estimate biomass and open up new applications. LCA also gives information on the
presence of large trees in a study area, which other metrics such as MCH cannot do. It is an
important point, considering that large trees are often the most affected by natural disturbance
and targeted by logging companies.
**Changes to manuscript**: ls.455-457: "LCA provides information on the presence of large trees
in a study area, which other metrics such as MCH cannot do. It is an important point, considering
that large trees are often the most affected by natural disturbance and targeted by logging
companies."
ls.564-565: "The comparison of LCA and MCH metrics showed that both performed similarly in
estimating AGB, highlighting the importance of large canopy trees to estimate biomass."
ls.645-647: "The results of our study may encourage further research in the use of Lidar data for
detecting the distribution of larger trees in tropical forests for ecological and conservation
studies."
**Comment:** The paper is framed around comparing the new LCA method against the existing
MCH method, but a clear comparison of the two against ground-based validation data is not
presented.
**Response**: Thank you for pointing this out. We added a short paragraph in the method section, as
well as a new section in the Results and in the Discussion, comparing the performance of LCA
and MCH methods. This is presented in the Methods (ls.218-240), in the Results (ls. 345-379)
and in the Discussion (ls.563-569).
To avoid any confusion, we moved the MCH local estimations of AGB from the main Lidar data
paragraph to the Supplementary information (S.2). $AGB_{Lidar}$ was also renamed $LCA_{Local}$ for
clarity.
**Changes to manuscript:** see ls. 218-240, ls. 345-379 and ls. 563-569. Figure 5 (attached here as
Fig. 1)
We chose to keep Table S3 in the Supplementary Information for clarity, but we added a figure
comparing AGB estimations using the 2 methods (Figure 5).
**Comment:** Is LCA quicker to calculate than MCH? It would be useful to present a comparison
of the computational time taken to calculate LCA versus MCH.
**Response:** LCA is not quicker to calculate than MCH, but it is not significantly slower either
(below 1s for both methods). Also, the strength of LCA lies in the structural information it
provides, not in its computational time. Thus, we chose not to add a detailed comparison of
computational time.
**Comment:** The application to detect the impacts of selective logging is potentially very
important.
**Response:** We agree. We emphasized this point in the Discussion:

**Changes to manuscript:** ls.609-611: "LCA could become an important tool to detect forest
degradation, in particular selective logging, considering that large trees are targeted by logging
companies."
**Comment:** My main suggestion to improve this paper are to concentrate on testing the relative
performance of LCA and MCH approaches at estimating biomass when validated against
inventory data (even if LCA performs worse, this is still a very useful result for method
development),
**Response:** Thank you for your suggestion. As mentioned above, we added a paragraph in the
method section, as well as two new sections (results and discussion) and a figure comparing the
two methods, showing that they perform very similarly. We also show how they differ in terms
of AGB estimations in different sites.
**Comment:** and comparing the performance of the two approaches when applied to detect the
impacts of selective logging.
**Response:** We compared the performance of the 2 approaches when applied to selective logging
detection. The MCH model showed a loss of biomass of 19 Mg ha$^{-1}$, compared to 15 with LCA
and 9 from a previous study based on rh25. We added this information in the results and the
discussion.
**Changes to manuscript:** ls.393-394: "As a comparison, the MCH model led to an estimated
biomass loss of 19 Mg ha$^{-1}$."
ls.607-609: "The higher biomass loss estimation from the MCH model (19 Mg ha$^{-1}$) again shows
how different metrics can lead to different results. Here, three methods based on three different
Lidar metrics yielded results that differed by more than twofold.".
**Comment:** I agree with reviewer 1 in that I don't see much value in testing the performance of
LCA against biomass estimates using MCH.
**Response:** Thank you for your suggestion. We removed Figure 5b. Performance comparison of
LCA and MCH model at the calibration sites is now based on Figure 5a. The models applied to
the nine sites are now Figure 5b, following your other suggestion to focus on the comparison of
LCA and MCH methods.
**Specific comments:**
**Comment:** Line 205 – How was bias calculated?
**Response:** We added the definition of bias to the manuscript:
**Changes to manuscript:** ls.214-215: "bias (mean difference between the expected values of
AGB and the observed values of AGB)".
**Comment:** Line 262 – What are the other models apart from a power law fit?
**Response:** For both LCA and MCH models, we tested linear models and power laws, which are

346 the 2 common fits. We modified the sentence to avoid any confusion:
347 **Changes to manuscript:** ls.302-303: "with a better coefficient of correlation and RMSE than a
348 power law fit"

350 **Comment:** Line 262 – 263 – Are RMSE values and r squared values here from cross-validation
351 or from the training data? Line 263 – Just present the bias from cross-validation.

353 **Response:** $R^2$ and RMSE are from training data.
354 We removed the bias from the training data and present the bias from cross-validation.
355 **Changes to manuscript:** l.304: "$bias_{cross\_val} = 0.16$ Mg"
356 ls.334-336: **"**coefficients of correlation, RMSE and bias from training data and cross-validation
357 are reported in Table 3."

359 **Comment:** Line 271 – How feasible is it to scale by wood density in the absence of inventory
360 data? Presumably errors would be larger if modelled estimates of wood density were used.

362 **Response:** We agree. If there is no information in the literature from previous studies, modelled
363 WD could be used, but would indeed give greater errors. This is now covered in the Discussion.
364 **Changes to manuscript:** ls.558-561: "In the absence of information on wood density from the
365 literature, modelled wood density could potentially be used, but would give greater errors. These
366 errors should be taken into account when reporting on the uncertainty of the results."

368 **Comment:** Lines 287-301 – It would be useful to also see how MCH performs at detecting this
369 loss of biomass.

371 **Response: The** MCH model (Table S3) gives a biomass loss of 19mg/ha, more than twice what
372 was reported in Andersen et al., 2014. These results were added to the results section and the
373 discussion section 4.6.:
374 **Changes to manuscript:** ls.393-394: "As a comparison, the MCH model led to an estimated
375 biomass loss of 19 Mg $ha^{-1}$."
376 ls.607-609: "The higher biomass loss estimation from the MCH model (19 Mg $ha^{-1}$) again shows
377 how different metrics can lead to different results. Here, three methods based on three different
378 Lidar metrics yielded results that differed by more than twofold.".

380 **Comment:** Lines 376-377 – This is a very nice approach to identify how much biomass is
381 missed by LCA.

383 **Response:** Thank you for this positive comment.

385 **Comment:** Figure S2 - Given that the minimum cluster size didn't have a major effect on the
386 AGB estimates, I would be interested in seeing a comparison of the performance of the LCA
387 metric just following masking versus the LCA metric following removal of segments below the
388 threshold cluster size. How computationally costly are these last steps?

390 **Response:** This is a good point. For a reference image of 1000x1000m pixels, the full process
391 takes less than one second. Just using masking may be slightly faster, but the computational cost is not an issue here. Just using masking gives similar results as when using LCA, because the
pixels removed by the full process represent a small fraction of the area covered by large trees
(1.73% on average). (R2=0.78, RMSE=45.7, bias=0.55)
These isolated pixels either represent single branches reaching above 27m or the tip of a tree
whose crown is mainly below 27m. Therefore, these pixels have no meaning in terms of our
LCA metric and do not represent large trees. This is why we chose to remove them. The goal of
our study is to show that large trees are sufficient to estimate AGB. We clarified this point in the
manuscript:
**Changes to manuscript:** ls.450-454: "Clusters smaller than 100 m$^2$ add only a small fraction
(1.7% on average) to LCA values across sites.  Including these clusters in LCA would not impact
the performance of the model (similar $R^2$, RMSE and bias) and would allow to skip the final
steps of the LCA retrieval (see Fig. S2).  However, since these pixels either represent single
branches reaching above 27m or the tip of a tree crown, they have no meaning in terms of our
LCA metric and do not represent large trees.".
**Comment:** Technical comments: Inconsistent approach to using capitals in section headings.
Line 209 – => Detecting changes of selective logging. Line 385 - => LCA as an AGB estimator
**Response:** Thank you for pointing this out. We removed the capital letters accordingly.

**Additional changes**

We made some additional minor edits to the paper to clarify some sentences. Please refer to the
track changes of the revised manuscript, notably:

-   Paragraph ls.485-503.
-   Figure 6: "2012" was replaced by "2011".
-   The word "nine" was removed from the title to be more consistent with the content of the
manuscript.

[revised manuscript text omitted]

<table>
<tr><td>**Deleted:** ,</td></tr>
<tr><td>**Deleted:** 5</td></tr>
<tr><td>**Deleted:** 3</td></tr>
<tr><td>**Deleted:** 5a</td></tr>
<tr><td>**Deleted:**</td></tr>
<tr><td>**Deleted:** b</td></tr>
<tr><td>**Deleted:** ,</td></tr>
<tr><td>**Deleted:** applied the model from Eq. (3) to all 1km$^2$ areas and</td></tr>
<tr><td>**Deleted:** the derived</td></tr>
<tr><td>**Deleted:** AGBLidar</td></tr>
<tr><td>**Deleted:** (see Sect. 2.2)</td></tr>
<tr><td>**Deleted:** , for which local models based on MCH were used</td></tr>
<tr><td>**Deleted:** cb</td></tr>
<tr><td>**Deleted:** . Global RMSE was found to be 34.72 Mg and RMSE per site varied between 20.79 Mg at BCI and 49.58 Mg at Manaus</td></tr>
<tr><td>**Deleted:** Our ground calibrated LCA model defined by Eq. (3) had a similar performance as the MCH based AGB model ($R^2_{MCH} = 0.79$, $RMSE_{MCH} = 44.2$ Mg, Table S3). These findings</td></tr>
<tr><td>**Deleted:** gives comparable results</td></tr>
<tr><td>**Deleted:** s</td></tr>
<tr><td>**Deleted:** , such as MCH</td></tr>
<tr><td>**Deleted:** ure 5b and 5c</td></tr>
</table>

[revised manuscript text omitted]

Figure 3

[Figure]

Figure 4

[Figure]

[Figure]

Figure 5

[Figure]

[Figure]

Figure 6

[Figure]

(a)

Δ LCA

-1                                    28

(b)

(c)    LCA (2010)

(d)    LCA (2011)

(e)    Δ LCA (2010 - 2011)

Disturbed
Recovered

LCA$_{2010}$   (mean = 34.8)

LCA$_{2011}$   (mean = 29.2)

Overlap

[Figure]

Figure 7

[Figure]

---

## Author Response (AR2)

Dear Editor,

Thank you for handling this manuscript. We addressed the technical corrections you suggested and we believe that the manuscript is now acceptable for publication.

Sincerely,

Victoria Meyer, on behalf of all co-authors.

**Technical corrections bg-2017-547**

**Abstract:**
**L 45: Please add a space between "1" and "ha"**
A space was added.

**Introduction:**
**L. 89: Should be "e.g."**
"eg." was replaced by "e.g."

**Material and Methods:**
**L. 117: Please insert a space between "2000" and "mm"**
**L. 146 and 154: Please add a apace between "1" and "m"**
**L. 164: Please insert space between "LCA" and "(h)"**
**L. 165/166: Please add space between number and unit (3 times)**
**L. 198: Please add space between number and percentage**
Spaces were added L.117, 146, 154, 164, 165, 166, 198
**L. 137: Delete comma after "Fearnside"**
The comma was deleted.

**Results:**
**L. 216: Please indicate a space between number and unit**
**L. 240 L. 165/166: Please add space between number and unit (2 times)**
**L. 244: Please add space between number and unit (2 times)**
**L. 283: Please indicate a space between number and unit**
**L. 284: Add a space before and after "="**
**L. 287: Add a space before and after "="**
Spaces were added L.216, 240, 244, 283, 284, 287
**L. 260: The abbreviation for Wood density WD should be either consistently used in the manuscript or avoided. Alternatively many studies use "ρ" to indicate wood density (as you did in the supplements), however it should be clearly defined in the text.**
The abbreviation WD is now consistently used throughout the manuscript and supplement.

**Discussion:**
**L. 313: Should be "e.g."**
"eg" was replaced by "e.g."
**L. 322 and 464: Add a comma after et al.**
Commas were added.
**L. 330: Sometime spaces are indicated after ">" or before "%", in other cases not, please use constantly the same format.**
Spaces were consistently added before ">" and "%" throughout the manuscript, figures and supplement.
**L. 362 and 367: Please indicate a space between number and unit**
Spaces were added.

**References:**
**L. 80c ", 2010."?**
", 2010" was removed.

**Tables and Figures:**
**Legend of Table 1: Indicates with uppercase letter. Please format the references in the legend and indicate the significance of "WD" and "AGB" in the legend or below the table as the reader can interpret the results without consulting the text of the manuscript. If you indicate "annual rainfall" there is no need to indicate "yr-1".**
"Indicates" now has uppercase letter. The references were formatted. The significance of WD and AGB were added in the legend. "yr-1" was removed.
**Table 1: It would be helpful to indicate the source of rainfall data. Is the annual rainfall in Chocó really 10 meters. I know it rains a lot in this region, but is it such high?**
The source of the rainfall data (WorldClim) was added and the reference added to the references list. We based our Choco rainfall number on available literature, but it is now matching the WorldClim data in the specific area we are studying. Chocó rainfall is now 6000mm.
**Legend of Table 3: Indicate (WD) after wood density in the legend**
"(WD)" was added.
**Legend of Figure 4: Shouldn´t it be average wood density (WD)?**
The sentence was corrected.
**Legend of Figure 5: Please add space between number and unit**
A space was added

**Supplement:**
**Legend of Figure S2: Please indicate "height" in lowercase letter**
"height is now in lowercase letter.
**L. 18: The reference should be Feldpausch. Delete comma after "2013"**
The name of the author was corrected and a comma was added.
**L. 34 and 126: Please indicate the "-" symbol after cm as subscripted symbol**
"-" is now a subscripted symbol.
**Table S1: I suggest to indicate the value of RMSE for Antimary also with two decimals**
Decimals (0) were added.
**Table S3: Also here I suggest to indicate two decimals for the values -0.3 (Bias) 0.8 (R2)**

Decimals (0) were added.

**Legend of Figure S4: Please insert a space between "1" and "ha" (L. 85)**

**Legend of Table S4: Please insert a space between "trees" and "≥50" (L. 85)**

Spaces were added to Figure S4 and Table S4.

**Figure S5: This figure should be improved as the titles of the x- and y-axes are difficult to read**

x- and y- axes labels were made bigger.

**The listed references should be formatted following the guidelines of Biogeosciences**

**Please correct:**

**Chave et al. 2005; Correct the coauthor´s last names (Riéra, Yamakura), substitute the comma after the title by a dot.**

**Zanne et al. 2009 Please indicate title of the journal, volume and pages**

References were corrected and formatted. There is no journal, volume and pages for the Zanne et al. reference.

**Additional changes:**

The affiliation of one of the co-authors has changed: Fernando Espírito-Santo,

[revised manuscript text omitted]

Antimary   BCI   Choco   La Selva   Manaus   Nouragues   Paracou   Tapajos   Cotriguaçu m m m m m

———— 1 km       ☐ Canopy Height > threshold , minimum 100 contiguous pixels

Figure 2

[Figure]

Figure 3

[Figure]

Figure 4

[Figure]

Figure 5

[Figure]

Figure 6

[Figure]

Figure 7

[Figure]

[Figure]

, 2010.